# Understanding greenhouse gas (GHG) column concentrations in Munich using WRF

Xinxu Zhao[1], Jia Chen[1], Julia Marshall[2], Michal Galkowski[3,4], Stephan Hachinger[5], Florian Dietrich[1], Ankit Shekhar[6], Johannes Gensheimer[1], Adrian Wenzel[1], and Christoph Gerbig[3]

[1]Electrical and Computer Engineering, Technische Unversität München, 80333 Munich, Germany
[2]Deutsches Zentrum für Luft- und Raumfahrt (DLR), Institut für Physik der Atmosphäre, Oberpfaffenhofen, Germany
[3]Max Planck Institute for Biogeochemistry, 07745 Jena, Germany
[4]AGH University of Science and Technology, Faculty of Physics and Applied Computer Science, Kraków, Poland
[5]Leibniz Supercomputing Centre (LRZ) of the Bavarian Academy of Sciences and Humanities, Boltzmannstr. 1, 85748 Garching, Germany
[6]Department of Environmental Systems Science, ETH Zurich, Universitätstrasse 2, 8092 Zürich, Switzerland

**Correspondence:** Jia Chen (jia.chen@tum.de)

**Abstract.** To address ambitious goals of carbon neutrality set at national and city scales, a number of atmospheric networks have been deployed to monitor greenhouse gas (GHG) concentrations in and around cities. To convert these measurements into estimates of emissions from cities, atmospheric models are used to simulate the transport of various trace gases and help interpret these measurements. We set up a modelling framework using the Weather Research and Forecasting (WRF) model applied at a high spatial resolution (up to 400 m) to simulate the atmospheric transport of GHGs and attempt a preliminary interpretation of the observations provided by the Munich Urban Carbon Column Network (MUCCnet). Building on previous analyses using similar measurements performed within a campaign for the city of Berlin and its surroundings (Zhao et al., 2019), our modelling framework has been improved regarding the initialization of tagged tracers, model settings, and input data. To assess the model performance, we validate the modelled output against two local weather stations and two radiosonde observations, as well as observed column GHG concentrations. The measurements were provided by the measurement campaign that was carried out from 1 to 30 August, 2018. The modelled wind matches well with the measurements from the weather stations, with wind speeds slightly overestimated. In general, the model is able to reproduce the measured slant column concentrations of $CH_4$ and their variability, while for $CO_2$, a difference in the slant column $CO_2$ of around 3.7 ppm is found in the model. This can be attributed to the initial and lateral boundary conditions used for the background tracer. Additional mismatches in the diurnal cycle could be explained by an underestimation of nocturnal respiration in the modelled $CO_2$ biogenic fluxes. The differential column method (DCM) has been applied to cancel out the influence from the background concentrations. We optimize its application by selecting suitable days on which the assumption of the DCM holds true: a relatively uniform air mass travels over the city, passing from an upwind site to a downwind site. In particular, STILT is used here and driven by our WRF modelled meteorological fields to obtain footprints (i.e., the potential areas of influence for signals observed at measurement stations), further used for interpreting measurement results. Combining these footprints with local knowledge of emission sources, we find evidence of $CH_4$ sources near Munich that are missing or underestimated in the emission inventory used. This demonstrates the potential of this data-model framework to constrain local sources and improve emission inventories.

# 1 Introduction

Human activities have resulted in an increase of the global average temperature relative to pre-industrial levels of approximately
1.1 °C, a number which is expected to reach around 1.5 °C between 2030 and 2052 (Masson-Delmotte et al., 2018; Allen
et al., 2018). To achieve the long-term goal of the Paris Agreement to limit temperature increase to 2 °C relative to pre-industrial levels, effective and efficient mitigation at national, regional and local levels is needed, leading to deep reductions in
atmospheric emissions of greenhouse gases (GHGs) over the coming decades (Masson-Delmotte et al., 2021). More than half
of the world's population resides in urban areas, which are directly responsible for over 30 % of the global GHG emissions
(Masson-Delmotte et al., 2018) and approximately 65 % of global energy use (IRENA, 2016). Thus, cities play a vital role in
addressing the challenge of carbon mitigation. The development of science-based methods to estimate carbon emissions from
urban areas is crucial for developing effective and coherent adaptation actions, and monitoring their success.

As the continent with the highest population density, Europe plays a major role in future mitigation efforts. In recognition
of this fact, the European Commission aims to make Europe climate neutral by 2050 (EU Commission, 2018). Furthermore,
member countries of the European Union (EU) have also adopted individual strategies consistent with that goal. For example,
the German government plans to reduce the national GHG emissions by more than 65% compared to 1990 by 2030, achieving
climate neutrality by 2045. Local-scale initiatives have also been put in place. Munich, currently the third largest city in
Germany with over 1.5 million inhabitants, has set an even more ambitious goal, aiming to be climate neutral by 2035.

To confront the challenge of carbon mitigation in cities and reach the goals set by individual municipalities, a multitude of
urban atmospheric networks have been built worldwide to optimize urban emissions (DeCola et al., 2018). Using the measurements they provide, GHG concentrations can thus be monitored in and around cities, and more accurate emission estimates can
be derived interpreting these measurements with atmospheric transport modelling and statistical techniques (Lauvaux et al.,
2016; Staufer et al., 2016). Based on these quantitative assessments, more reliable scientific guidance can be provided to
policymakers in order to plan local emission reductions effectively and monitor mitigation efforts.

Two standard approaches are widely used for estimating emission fluxes: the bottom-up approach and the top-down approach. Using the bottom-up approach, the total fluxes are estimated on the basis of statistical activity data from individual sectors (e.g., power plants, traffic) and the corresponding emission factors. This approach is widely used for generating
global and national sector-by-sector emission inventories, e.g., the Emissions Database for Global Atmospheric Research
(EDGAR; Janssens-Maenhout et al. (2019)). The emissions produced using this technique are often quite uncertain, owing
to missing or simplified knowledge of emission processes and the considerable heterogeneity in space and time (Klausner
et al., 2020). By using the top-down approach, estimated emission fluxes can be refined using additional information provided
by measurements of atmospheric concentrations. Prior emission estimates, usually estimated by the bottom-up approach, are
used as inputs for an atmospheric transport model and the resultant concentrations are compared to the atmospheric composition measurements (Zhao et al., 2019; Shekhar et al., 2020). The emission estimates can then be optimized using either a
mass-balance approach (Heimburger et al., 2017) or other inverse techniques (Jones et al., 2021). In terms of GHG emission
estimation for cities or an area of interest, the top-down inversion approach has frequently been applied in modelling studies

accompanied by urban measurement networks, e.g., in California (Turner et al., 2016), Paris (Staufer et al., 2016), Boston (Sargent et al., 2018), Berlin (Klausner et al., 2020) and Indianapolis (Jones et al., 2021). Inversion models still show considerable potential for improvement, owing to limited knowledge about the characteristics and spatial distribution of emission sources

(e.g., missing or underestimated sources, inner-city traffic), uncertainties in background concentrations, and the difficulty of modelling transport in complex urban environments, and thus representing the observations. Furthermore, emissions that are highly heterogeneous in time and space are challenging from the perspective of both measurements and modelling, and demand extra care in their interpretation (e.g., Vaughn et al. (2018)).

  To aid in reaching the goal of climate neutrality and track emissions in Munich, our group has established a novel automated
urban sensor network (MUCCnet: Munich Urban Carbon Column Network, accessible via http://atmosphere.ei.tum.de/, Dietrich et al. (2021)) for continuous, long-term monitoring of GHGs in and around Munich. In brief, MUCCnet takes measurements simultaneously at five locations to capture concentration signals in and around the city, with the goal of estimating city emissions using continuous measurements accompanied by atmospheric models. MUCCnet is designed with the differential column methodology (DCM; Chen et al. (2016)) in mind, developed to quantify the emissions within a certain area, e.g., a
single city. This is done by capturing concentration enhancements between downwind and upwind sites, so that a signal can be attributed to the emissions from the area in between. Several studies have used this approach combined with atmospheric models to study urban and local emissions, e.g., in Berlin (Hase et al., 2015; Zhao et al., 2019), Paris (Vogel et al., 2019), Munich (Toja-Silva et al., 2017) and Chino, California (Chen et al., 2016; Viatte et al., 2017). Detailed descriptions of MUCCnet and its measurement principle are presented in Sect. 4.1 and Dietrich et al. (2021).

On the basis of MUCCnet measurements and our previous study that interpreted data from a measurement campaign around Berlin (Hase et al., 2015; Zhao et al., 2019), we have set up a modelling framework for Munich. This is based on the Weather Research and Forecasting model (WRF) enhanced with a biospheric flux module (WRF-GHG; Beck et al. (2012)), which runs at a horizontal resolution of up to 400 m over the city. WRF is a mesoscale model commonly used for weather and atmospheric studies, and WRF-Chem has been extended with additional modules for tracer transport and chemistry (now including the
GHG modules; Skamarock et al. (2008)). With WRF-Chem (Peckham et al., 2017), modelled meteorological fields are used to drive simulations of the atmospheric transport of trace gases, e.g., GHGs (Zhao et al., 2019) and air pollutants (Georgiou et al., 2018). Our model for Munich aims to reproduce observations from the five measurement locations in MUCCnet so as to aid in their interpretation and better understand the processes driving the emission and uptake of GHGs around the city. Apart from generating concentration fields for different emission tracers, the output from the modelling system will be further used
as input for other studies. For instance, highly-resolved meteorological fields can drive particle transport models (Fasoli et al., 2018). These Lagrangian footprints can then be used for inversion studies, similar to Heerah et al. (2021), who optimized dairy $CH_4$ emissions across the San Joaquin Valley using WRF coupled with Stochastic Time-Inverted Lagrangian Transport model (WRF-STILT) inversions. Currently, an adapted Bayesian inversion model based on Jones et al. (2021) is being developed to infer anthropogenic $CO_2$ emissions, with the consideration of biogenic fluxes.

In this paper, we describe our model framework in detail and apply it in the interpretation of the observations collected by MUCCnet from 1 to 30 August, 2018. Compared to the previous study in Berlin (Zhao et al., 2019), the model has been

updated and several aspects related to model settings and initialization processes have been improved. As an example, more precise anthropogenic fluxes have been used for the tagged emission tracers. All these model-related aspects are described in Sect. 2. In Sect. 3, we assess the performance of our model by comparing its output to the measurements from two local weather stations. A model-measurement comparison of GHG column concentrations is presented in Sect. 4. In Sect. 5, we optimize the application of DCM, and use it to assess the model performance and further track missing or underestimated emission sources around MUCCnet, through combining footprints generated by the particle transport model STILT (Fasoli et al., 2018) and the knowledge of local sources.

## 2 Modelling Framework Description

We use WRF-Chem Version 3.9.1.1 (Skamarock et al., 2008; Peckham et al., 2017) with an updated GHG module (Beck et al., 2012) in order to simulate the fluxes and transport processes of atmospheric GHGs in and around Munich at a horizontal resolution of up to 400 m. The main component of the setup is the WRF-ARW model, based on the fully compressible non-hydrostatic Euler equations (Skamarock et al., 2008). We take realistic meteorological driving data from the ERA5 reanalysis (Hersbach et al., 2020), extracted at approximately 31 km horizontal resolution and with 137 vertical levels from ground level to 0.01 hPa. These data provide the initial and boundary conditions for the meteorological fields.

In our model, biogenic fluxes of GHGs are simulated online, driven by meteorological parameters at native model resolution, in addition to other inputs. Furthermore, fluxes from external emission inventories are included as inputs and these surface fluxes are transported as passive tracers in WRF (Beck et al., 2012). Compared to the originally developed WRF-GHG, in which the GHG modules had to be explicitly integrated with WRF Version 3.2 (cf. Zhao et al., 2019), these modules have been added to the official WRF-Chem repository since WRF-Chem Version 3.4. It is worth noting that the GHG module does not take into account atmospheric chemical reactions (as it treats GHGs as passive tracers). This is, however, not expected to produce significant biases, owing to the long lifetimes of GHGs compared to the relatively short residence time of tracers in the regional domain (Super et al., 2016; Dekker et al., 2017).

The model is configured in a three-domain nested configuration, with horizontal resolutions of 10 km for the outermost domain (D01), 2 km for the intermediate domain (D02) and 400 m for the innermost domain (D03), as illustrated in Fig. 1. The spatial grids are assigned using the Lambert Conformal Conic (LCC) projection. The simulations are carried out with model integration time steps of 30 seconds, 6 seconds 1.2 seconds for each domain, with model outputs saved at time intervals of 3 hours, 1 hour, and 15 minutes, respectively. We define 46 vertical levels from the surface up to 50 hPa, 21 of which are in the lowest 1 km of the atmosphere. All five total column measurement sites from MUCCnet are located inside D03. The Mellor-Yamada-Janjic (MYJ) planetary boundary layer (PBL) parameterization scheme is employed to resolve modelled vertical turbulent mixing and accurately depict meteorology conditions (Hu et al., 2010).

To better capture the urban landscape features and improve the urban model performance (Ching et al., 2018; Mughal, 2020), extra urban land use land cover categories are provided for the innermost domain (D03, area of Munich), which enables us to use the urban canopy multi-layer scheme in WRF (Brousse et al., 2016). This is done by re-classifying the land-cover

categories for urban areas, while keeping the other land-cover categories unmodified. The re-categorized land-cover types are derived from the European Local Climate Zones (LCZ) map (Demuzere et al., 2019), extracted for our high resolution domain (Fig. 1). More information regarding this procedure can be found in Sect. S3 of the supplement.

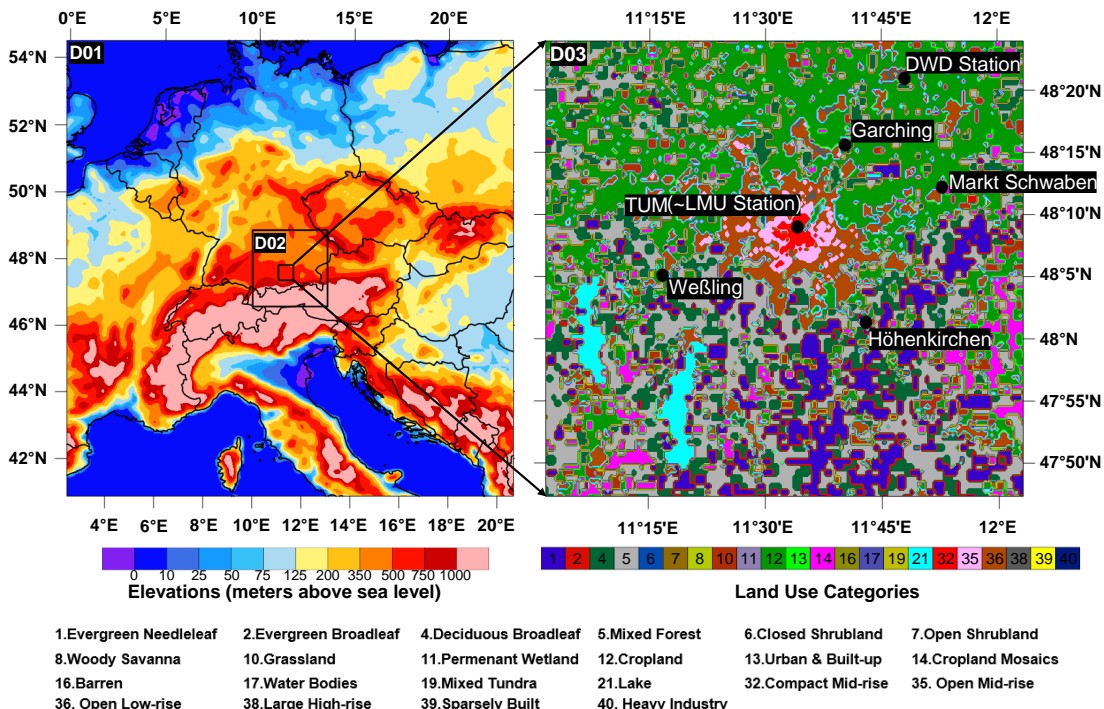

**Figure 1.** Topography map for the entire domain area (left panel). The right panel shows the land use classification in D03, including the 16 IGBP-modified MODIS land-cover types (from 1 to 21, as illustrated in the color bar and labels), and 6 classified LCZ land-cover categories defined for the urban areas of D03 (numbers larger than 30, as illustrated in the color bar and labels). The five measurement sites in our MUCCnet campaign and the surface weather stations used for the model-measurement comparison of meteorological fields are marked as black dots on the right panel. The national boundaries and coastlines in the left panel are from National Center for Atmospheric Research (NCAR) Graphics Version 4.1 (© UCAR/NCAR).

The initial and lateral boundary conditions in the simulated background concentration fields of $CO_2$ and $CH_4$ are taken from the Integrated Forecasting System (IFS) Cycle 45r1, implemented by ECMWF, at a horizontal resolution of approximately 130 40 km (Rémy et al., 2019; Browne et al., 2019) The IFS Cycle 45rl is operated by the European Center for Medium-Range Weather Forecasts (ECMWF) as part of the Copernicus Atmosphere Monitoring Service (CAMS). The IFS cycle 45rl is referred to as CAMS for simplicity.

Biogenic $CO_2$ fluxes are implemented online utilizing the Vegetation Photosynthesis and Respiration Model (VPRM; Mahadevan et al. (2008); Chen et al. (2020)), a simple diagnostic light-use-efficiency (LUE) model coupled to WRF-Chem. VPRM does not reproduce the physiological processes of vegetation, but rather calculates Gross Primary Production (GPP) using the input of meteorological variables and vegetation indices derived from remote sensing data. Ecosystem respiration (RES) is estimated using a simple linear model related to the air temperature and vegetation-specific parameters. Finally, the hourly $CO_2$ Net Ecosystem Exchange (NEE) is the difference between GPP and RES. In detail, the entire calculation is based on satellite-derived indices, short wave radiation and surface temperature at 2 m above the ground level as simulated by WRF (Beck et al., 2012). The indices (i.e., Enhanced Vegetation Index (EVI) and Land Surface Water Index (LSWI)) here are derived from reflectance data measured by MODIS, specifically the product MOD09A1 Version 6. MODIS has resolutions of 0.5-1 km depending on the wavelength band and a temporal resolution of eight days (Vermote, 2015). The MODIS reflectance data are aggregated and interpolated onto the LCC projection, and the vegetation is classified following the SynerCover Product (SYNMAP) data with a resolution of 1 km (Jung et al., 2006).

Because the 1-km resolution of the SYNMAP dataset is unable to resolve vegetation within cities, urban areas are essentially masked out and VPRM does not produce any fluxes within cities (see the middle panel in Fig. 2). Thus, we extend and refine the vegetation classification using the Dynamic Land Cover map of the Copernicus Global Land Service at a resolution of 100 m (CGLS-LC100). This refined classification is used for our innermost domain to better capture the urban biogenic signals of $CO_2$. A comparative visualisation of the vegetation classification maps with and without the refinement around Munich is shown in Fig. 2. Details of this reclassification are described in Table S1 of the supplement.

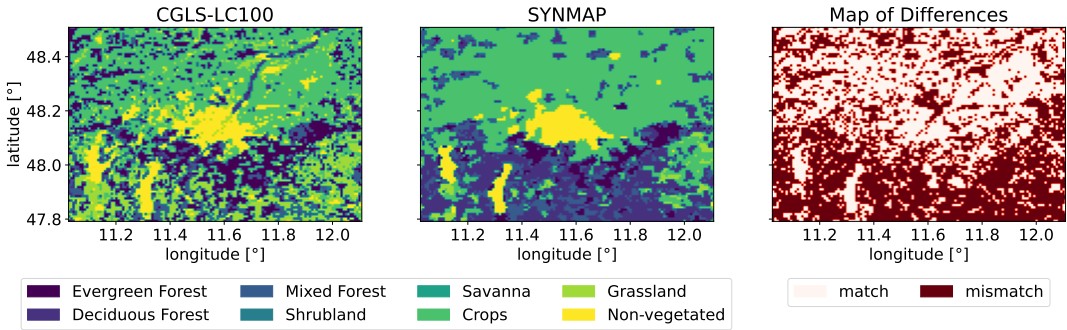

**Figure 2.** The maps of the vegetation classification with (left) and without (middle) refining using CGLS-LC100, and their difference (right).

$CH_4$ fluxes from wetlands are estimated using the Kaplan model (Kaplan et al., 2006), which is run online within WRF-Chem. This model calculates $CH_4$ emissions from anaerobic microbial production in wetlands as a fraction of heterotrophic respiration (Beck et al., 2013). The flux estimates depend on the modelled soil moisture, soil temperature and the carbon pool from the Lund-Potsdam-Jena model, which is used for classifying the wetland fractions in the domain (Beck et al., 2012).

The first version of the GHG and co-emitted species emission database produced by the Netherlands Organisation for Applied Scientific Research (TNO; dataset TNO_GHGco_v1.1; Super et al. (2020)) is used to initialize anthropogenic fluxes as tagged tracers. This dataset provides annual gridded anthropogenic emissions for 2015 at a horizontal resolution of 0.05 ° ×

0.1 ° (latitude × longitude, approximately 6 km × 6 km), covering most of Europe and part of North Africa. In addition to the gridded emissions, point emission sources (e.g., power plants) are reported separately with geographical coordinates. These emissions are classified into fourteen sectors, following the Gridding Nomenclature for Reporting (GNFR) emission categories (A to L), i.e., industry, public power, road transport and other an thropogenic sources. Furthermore, a high-resolution version of TNO_GHGco_v1.1, at a resolution of 1/120 ° × 1/60 ° (latitude × longitude, approximately 1 km × 1 km; van der Gon et al. (2019)) is available for central Europe (i.e., all of Germany and parts of France, Poland and the Netherlands, etc.). This version of TNO_GHGco_v1.1 was prepared to support model studies at the local scale in the $CO_2$ Human Emissions project (CHE; van der Gon et al. (2019)). Since our outermost domain is not fully covered by the high-resolution version of TNO_GHGco_v1.1, we use the lower resolution (6 km) emissions for the outermost domain (D01), and the high-resolution version for the other two domains (D02 and D03 in Fig.1). To prepare the input for WRF-Chem, the required temporal dis-aggregation of the annual emissions was performed based on time-dependent scaling factors for monthly, weekly, and diur-nal variations (Zhao et al., 2019; Super et al., 2020). In addition, we release the emission fluxes from the point sources in TNO_GHGco_v1.1 from different heights above the ground, using the vertical profiles provided in Table 2 of Brunner et al. (2019). The re-allocation of point sources to vertical levels in our domain is illustrated in Fig. S2 of the supplement.

## 3 Model-measurement comparison for wind fields

To assess the performance of our model framework and evaluate the modelled meteorological variables used for transporting the fluxes, some key meteorological parameters are compared to measured values provided by two local weather stations and two radiosonde observations.

### 3.1 Comparison for surface wind fields

The first station is located at the Meteorological Institute of the Ludwig Maximilian University of Munich (LMU; latitude: 48.15 °, longitude: 11.57 °, altitude: 561 m), close to the center site of MUCCnet. This station can provide time series of meteorological variables second by second. We compare the model to five meteorological variables measured at LMU: the temperatures at heights of 2 m and 30 m above the ground (T2 & T30), the precipitation, and the wind speed and direction at 30 m above the ground (WS30 & WD30). The other station we consider is operated by the German Meteorological Service (Deutscher Wetterdienst in German, DWD). This automated weather station is located at the Munich airport and has station ID 01262 (latitude: 48.35 °, longitude: 11.81 °, altitude: 446 m). We use the following variables measured at the airport for comparison with our model: T2, precipitation, relative humidity, air pressure, and the wind speed and direction at 10 m above the ground (WS10 & WD10).

As one of the key drivers for the transport of trace gases in the model, the simulated wind field directly impacts the transport patterns of the tracers. Thus, it is particularly important to assess the model performance with regards to the wind field. Here, we employ the measured WS30 & WD30 at the LMU station, and WS10 & WD10 at the DWD station. The LMU station measures the winds every second, while the wind data given by DWD are recorded as 10-minute means. We apply a cut-off

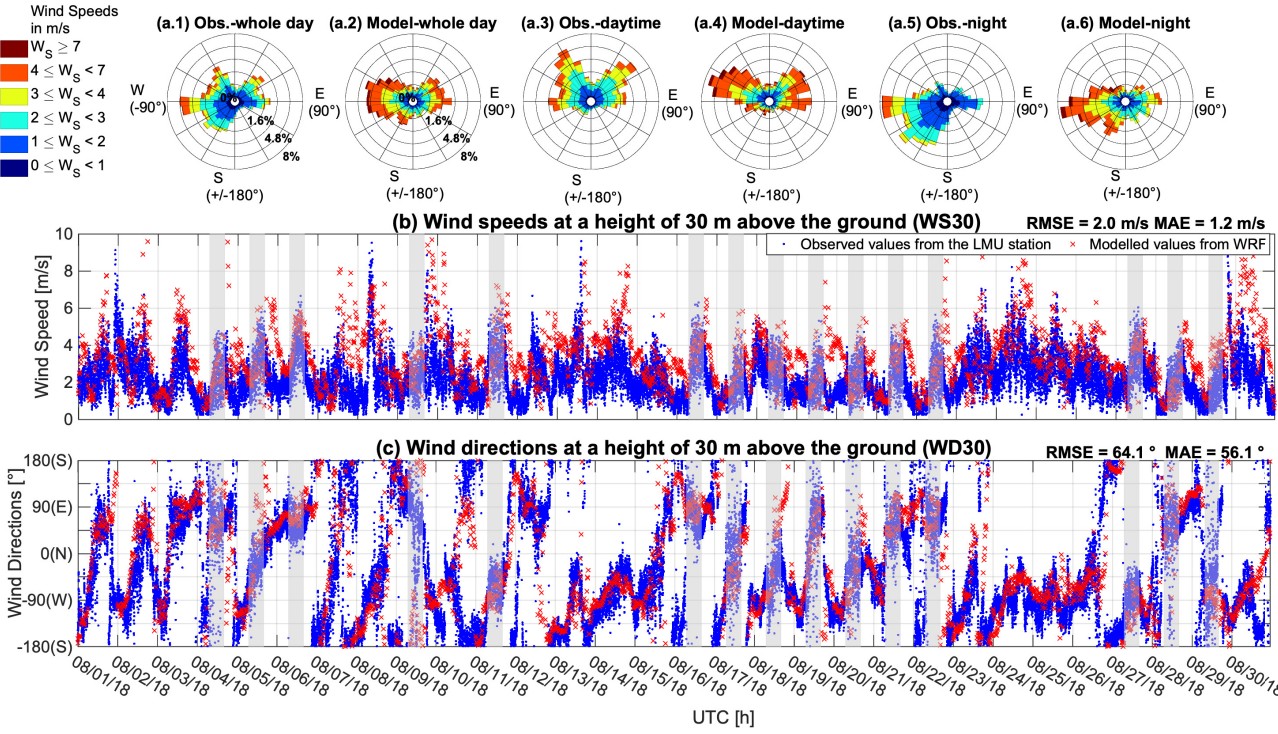

**Figure 3.** Wind roses (a) and time series of simulated and observed wind speeds (b) and wind directions (c) at a height of 30 m above the ground at LMU. Wind roses for the measurements over August 2018 are plotted in Panels (a.1): 24 hours, (a.3): daytime from 6 am to 5 pm only and, (a.5): nighttime only, while Panels (a.2), (a.4), and (a.6) represent the modelled values. Each wind rose indicates WS30, WD30 and the frequency (% scale) of wind coming from a particular direction during the targeted period. The blue dots in Panels (b) & (c) represent the measured values from the LMU station and the red crosses represent the simulation. The grey shaded areas mark the measurement periods used for the model-measurement comparison of column concentrations in Sect. 4.

wind speed threshold (0.5 m/s in our case) to the values shown in Fig. 3, owing to large uncertainties in wind directions during low wind speed periods (Zhao et al., 2019).

A comparison between the modelled and measured winds at the LMU station is shown in Fig. 3. Prevailing wind directions both in the simulations and measurements are either easterly or westerly during the daytime, while the prevailing winds at night are generally from the southwest. The measurements (panels (a.1), (a.3) and (a.5)) show larger scatter in the wind direction over

August compared to the simulations. The evaluation of wind directions are treated following the method presented in Jiménez and Dudhia (2013). Along the time series, the simulated (Fig. 3(b): red crosses) and measured (blue dots) wind speeds show similar variability, but the model generally overestimates wind speeds with a root mean squared error (RMSE) of 2.0 m/s and a mean absolute error (MAE) of 1.2 m/s. During the measurement periods (cf. Sect. 4) by the grey shaded areas in Fig. 3(b), the model performs better with a RMSE of 1.6 m/s and a MAE of 1.1 m/s. Regarding a comparison of the wind directions

between the model and measurements (see Fig. 3(c)), the model mostly follows the measured fluctuations of wind directions but with some difference over time (RMSE = 64.1 ° & MAE = 56.1 °). The model performance is reduced in some periods, e.g., between 24 and 26 August, when the variability of the wind direction is remarkably lower in the model. Over the measurement periods marked by grey areas, smaller differences between the WRF and the in situ surface wind directions are found, with a RMSE of 58.2 ° and MAE of 51.8 ° .

In addition to this model-measurement comparison for WS30 & WD30, similar comparisons regarding the other meteorological variables have also been performed and are presented in Sect. S5 of the supplement. These comparisons indicate that our model has the capability to provide reasonable simulated meteorological fields for driving the transport of trace gases.

## 3.2  Comparison for vertical wind profiles

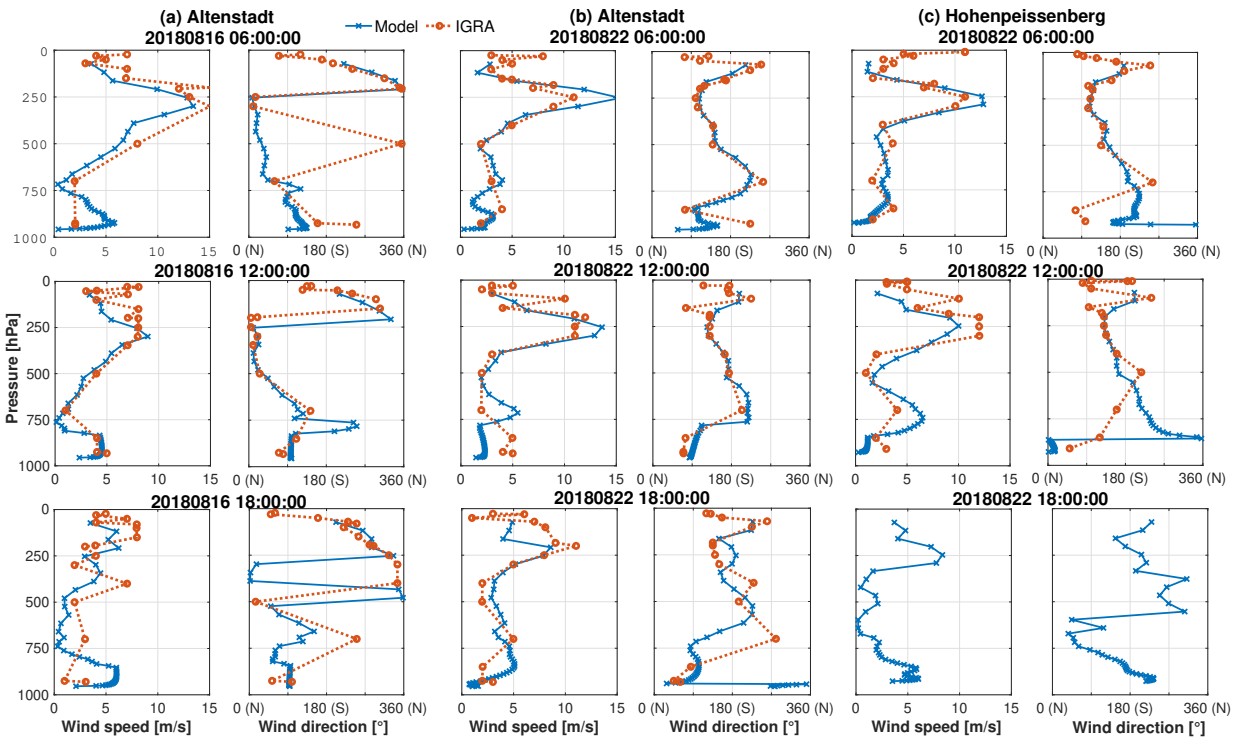

**Figure 4.** Wind profile comparison between the model (orange dashed line) and the measurements (blue solid line) from two radiosonde locations, i.e., (a) & (b) Altenstadt on 16 and 22 August, and (c) Hohenpeißenberg on 22 August. Measurements are lack in the IGRA at 18:00 on 22 August.

In order to assess whether the WRF meteorological fields allow the transport model (STILT; Fasoli et al. (2018)) to produce
realistic footprints (cf. Sect. 5.2), we have evaluated our modelled height-dependent wind fields using observations. This assesses their accuracy in the vertical dimension more deeply than the comparisons of the surface wind fields shown in Sect. 3.1.

Figure 4 shows modelled and measured wind speed and direction profiles for 16 and 22 August. The comparison uses radiosonde data from two sites (Altenstadt and Hohenpeißenberg, marked as black squares in Fig. 9(b)), provided by Integrated Global Radiosonde Archive (IGRA). This archive collects radiosonde and pilot balloon observations along significant vertical
levels historically and in near-real-time from around 800 distributed stations worldwide (Durre et al., 2018). On both days at Altenstadt (Fig. 4(a) & (b)), the model reproduces the observed winds well, especially at noon, while outliers in wind direction always exist at the lowest level in the morning and some mismatches appear at 18:00 UTC. Regarding the wind profiles measured at Hohenpeißenberg, the modelled wind directions at higher altitudes agree quite well with the observations, with mismatches close to the ground level. It is worth noting that our domain is close to the Alps, with complex topography making
meteorological modelling more challenging. The results need to be interpreted in this context.

## 4 Model-measurement comparison for concentration fields and model tracer analysis

This part of our study is dedicated to a comparison of the measurements from MUCCnet to column concentrations for different tagged tracers extracted from the simulation output fields. Section 4.1 contains relevant information on the measurements used for interpreting the model, and the model-measurement comparison is discussed in Sect. 4.2. The final section (Sect. 4.3)
characterizes changes in concentrations caused by individual anthropogenic emission processes, as represented by our model.

### 4.1 Description of MUCCnet and the measurement campaign

Our WRF model framework for Munich is designed to study GHG concentrations in connection with the Munich Urban Carbon Column network (MUCCnet; Dietrich et al. (2021)). In MUCCnet, five compact Fourier-transform infrared (FTIR) Spectrometers (EM27/SUN by Bruker Optics) have been deployed, four of which were located around Munich at a radius of 20 km for
the 2018 period analysed in this study (this was later changed to 10 km in 2019). The fifth instrument has been set up close to the center of Munich, at the TUM campus (see Fig. 1). By using the sun as a light source, the EM27/SUN measures near-infrared solar spectra (Gisi et al., 2012) In MUCCnet, the recorded interferograms are automatically transformed to spectra, converted to column averaged dry-air mole fractions (DMF) of $CO_2$ and $CH_4$ between the instrument and the end of the atmosphere in the direction towards the sun, and further uploaded to the official website of MUCCnet (https://atmosphere.ei.tum.de).
The retrieval algorithm GFIT GGG-2014 (Wunch et al., 2015) was applied during the measurement campaign of 2018, while currently, MUCCnet is using the PROFFIT (Hase et al., 2004; Frey et al., 2015, 2019; Alberti et al., 2022) and GGG-2020 algorithms (Laughner et al., 2023). Additionally, the hydrogen fluoride (HF) correction (Saad et al., 2014) is not applied in our retrieval process of $CH_4$. All five instruments are automatically operated and controlled using our universal enclosure systems and two software programs (Heinle and Chen, 2018; Dietrich et al., 2021). Detailed information on the EM27/SUN instrument
can be found, e.g., in Hedelius et al. (2016), Chen et al. (2016), Hase et al. (2016) and Frey et al. (2019).

In this study, we compare simulations to measurements collected during a campaign that was carried out from 1 to 30 August, 2018 (Dietrich et al., 2019, 2023). Table S4 in the supplement shows relevant parameters for assessing the measurement performance during that period, including the number of observations per day for each site and the ground-level wind infor-

mation for each day, i.e., the daily mean of WS30 & the approximate change in WD30 during the day, provided by the LMU station.

The EM27/SUN measures column-averaged DMF of $CO_2$ and $CH_4$, hereafter referred to as $XCO_2$ and $XCH_4$. Over the entire campaign period in 2018, the mean of the measured $XCO_2$ for all five sites is 404.4 ppm with a standard deviation of 1.2 ppm, ranging from 400.8 ppm to 408.1 ppm. For $XCH_4$, the measurements range from 1840.5 ppb to 1896.0 ppb, with a mean of 1865.5 ppb and a standard deviation of 9.1 ppb. Since the operation of the instruments is strongly influenced by weather conditions, the spatial and temporal measurement coverage for some days (e.g., 1-3 August) is limited (see Table S4). By assessing the main characteristics of the measurement days during the campaign, we selected fifteen days in total with good measurement conditions (i.e., with a quality level better than '++', cf. Table S4 and Sect. S6 of the supplement) to make the model-measurement comparison: 4-6, 9, 11, 16-22 and 27-29 August, 2018. Details of the campaign and side-by-side calibrations are discussed in Sect. 4.1 and 5 of Dietrich et al. (2021).

## 4.2 Model-measurement comparison of $XCO_2$ and $XCH_4$

In our modelling framework, the anthropogenic emission fluxes from an inventory (TNO-GHGco, see Sect. 2 for details) are used as fluxes for tagged tracers, each representing a source category group available in the inventory (Super et al., 2020). These tracers are transported passively throughout the model domains using internal WRF-Chem transport schemes. Then, the total concentrations for a trace gas are derived by summing up the contributions from individual emission processes (i.e., the different tagged tracers) and the background concentrations for this gas, provided by CAMS and advected from the model boundaries as a separate tracer.

### 4.2.1 Calculation of smoothed slant column concentrations

The modelled vertical concentration profiles are converted to pressure-weighted column-averaged concentrations. That is to say, for the trace gas $G$, the simulated column-concentration at a specific location and time $XG(x,y,t)$ can be calculated as follows:

$$XG(x,y,t) = \sum_{l=1}^{L_{ver}} [w_l(x,y,t) \times G_l(x,y,t)] \tag{1}$$

where $G_l$ stands for the simulated mole fraction at the location $(x,y)$ and time $t$ at the $l^{th}$ vertical layer of WRF. $L_{ver}$ is the total number of the vertical layers (i.e., 45 in our study) and $w_l$ is the weight of the $l^{th}$ vertical layer which can be obtained as:

$$w_l(x,y,t) = \frac{\Delta P_l(x,y,t)}{P_{sf}(x,y,t) - P_{tp}} \tag{2}$$

where $P_{tp}$ is the hydrostatic pressure at the top of the model (i.e., 50 hPa) and $P_{sf}$ is the surface pressure. $\Delta P_l$ denotes the pressure difference between the top and the bottom of the $l^{th}$ vertical layer. Here we use the WRF meteorology derived pressure profiles to calculate the weights for each vertical layer, instead of the pressure profiles used for EM27 retrievals (NCEP). This is convenient, as the WRF modelled profiles for both pressure and concentrations have the same vertical structure and no

interpolation is necessary. Furthermore, only slight differences in pressure at higher altitudes was found between the WRF
model and NCEP (see Sect. S10).

However, when comparing our modeled values to the measurements here, we need to consider the characteristics of the
instruments in a more accurate way. The EM27/SUN records the spectra along a slant column from the sun to the ground,
instead of a vertical column perpendicular to the ground. Simulated concentration fields of $CO_2$ and $CH_4$ used for the model-
measurement comparison in this study must therefore be aggregated along the slant columns from the ground to the sun during
the period of the available measurement dates.

In addition, when reconstructing the vertical structure of the atmosphere during the retrieval process of ground-based remote
sensing instruments (e.g., Vogel et al. (2019); Zhao et al. (2019)) and satellites (e.g., Ohyama et al. (2020)), an averaging kernel
(AK) is used to represent the altitude-dependent column sensitivity (Borsdorff et al., 2014). For solar-viewing instruments, its
shape is strongly dependent on the solar zenith angle (SZA). The retrieved quantity then typically depends considerably on
the AK as well as on the a-priori profile used in the retrieval. Accordingly, when the modelled values are compared to such
measurements, they also need to be smoothed using the AK and the a-priori profile in order for the comparison to be valid.
Since the EM27/SUN has a spectral resolution of only $0.5\,\mathrm{cm}^{-1}$, we can use a fitted AK matrix, which is obtained by applying a
simple least-squares fit to a given a-priori AK profile. The details related to applying the AK to the model data were previously
described in detail in Sect. 3.3 of Zhao et al. (2019).

Thus, the simulations are mostly interpreted by comparing the aggregated and AK-smoothed modelled values to column
measurements. The smoothed slant column concentration for a target gas $XG_{\mathrm{sla}}^{\mathrm{S}}$ is calculated following Eq. 3 (cf. Vogel et al.
(2019); Zhao et al. (2019)),

$$XG_{\mathrm{sla}}^{\mathrm{S}}(x,y,t) = \sum_{l=1}^{L_{\mathrm{ver}}} \mathrm{w}_{\mathrm{sla},l}(x,y,t) \times [\mathrm{AK}_{G,l}(t) \times G_{\mathrm{sla},l}(x,y,t) + (1 - \mathrm{AK}_{G,l}(t)) \times G_{\mathrm{pri},l}] \tag{3}$$

where $G_{\mathrm{sla},l}$ are modelled concentrations for trace gas $G$ at the $l^{th}$ vertical layer following the slant column along the line of
the sun, $\mathrm{AK}_{G,l}$ is the fitted AK of the gas $G$ at the $l^{th}$ vertical layer and time $t$, $G_{\mathrm{pri},l}$ stands for the mixing ratio of the a-priori
profile for trace gas $G$ at the $l^{th}$ vertical layer, and $\mathrm{w}_{\mathrm{sla},l}$ is the weight of the $l^{th}$ vertical layer along the slant column. We have
used the a-priori profile from the Whole Atmosphere Community Climate Model (WACCM) Version 6 for 2018.

It should also be noted here that the measured samples are filtered during the autonomous retrieval process in MUCCnet (Di-
etrich et al., 2021). Specifically, to reduce uncertainties caused by high air masses, measurements are discarded when they are
observed at SZA larger than 75 degrees (Tu et al., 2020; Gisi et al., 2012). Thus, the measurement period each day ranges
from around 6:00 UTC to 17:00 UTC and lasts for approximately 11 hours in summer. When comparing the simulations to
the measurements, the simulated $CO_2$ and $CH_4$ concentration profiles along the slant column during the period of the available
measurement dates are aggregated to the AK-smoothed column concentrations ($XCO_{2,\mathrm{sla}}^{\mathrm{S}}$ and $XCH_{4,\mathrm{sla}}^{\mathrm{S}}$) by using Eq. 3. In
the following figures, the grey shaded areas are used to mark these measurement periods used for the model-measurement
comparisons.

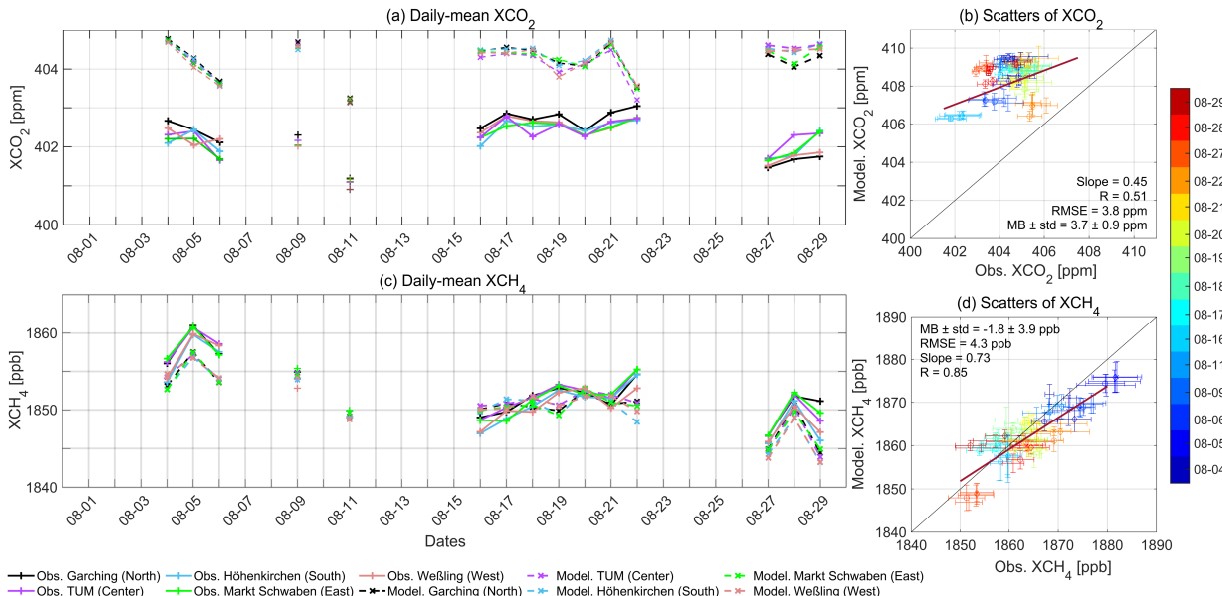

**Figure 5.** Time series and scatter plots for $XCO_2$ (a & b) and $XCH_4$ (c & d). In panels (a & c), the dashed lines represent the daily mean modelled $XCO_{2,sla}^S$ and $XCH_{4,sla}^S$, while the solid lines denote the measurements. Colors in panels (a & c) mark the different measurement sites. In scatter plots (panels (b & d)), colors represent the values for different measurement days, as marked in the color bar. The error bar represents the standard deviation of the measured and simulated values at each site.

### 4.2.2 Comparison of daily mean concentrations and estimation of $CO_2$ mean bias

The daily mean measured and modelled values ($XCO_{2,sla}^S$ and $XCH_{4,sla}^S$) for the 15 studied days and their scatter plots are shown in Fig. 5. When producing daily-averaged modelled values, we have considered the limited measurement period on each day, that is, from around 6:00 UTC to 17:00 UTC. For $CO_2$, the simulated smoothed column concentrations ($XCO_{2,sla}^S$, dotted

lines in Fig. 5(a)) are overall overestimated compared to the measurements, with a mean bias (MB) of 3.7 ppm $\pm$ 0.9 ppm, the latter value giving the standard deviation of the MB over all measurement days. This bias for $XCO_{2,sla}^S$ is mainly attributed to the initial and boundary conditions of the concentration fields in the model as provided by CAMS, which has also been seen in other studies. A discussion of the $XCO_2$ bias is included in Sect. S14 of the supplement. Gałkowski et al. (2021) found a similar bias between the CAMS product and airborne measurements in the free troposphere over Europe, with a MB of 3.7 $\pm$

1.5 ppm, which they have attributed to the far-field contributions to the local signal. While the agreement between the biases reported in their study and here is excellent, it should be noted that the numbers are not fully comparable, as the authors of the quoted study evaluated a limited section of the vertical column (namely between 3 – 10 km a.m.s.l.) using in-situ data, thus excluding the lower tropospheric (below 3 km) and the stratospheric components of the total column. Moreover, Tu et al. (2020) also reported a bias when comparing CAMS to their column measurements from the Collaborative Carbon Column

Observing Network (COCCON) site at Kiruna, Sweden, with a MB of 3.7 $\pm$ 1.8 ppm.

As can be seen in the scatter plot of $XCO_{2,\text{sla}}^{S}$ (Fig. 5(b)), the measurements generally exhibit more scatter (seen in the magnitude of the error bars in the x-direction) compared to the simulation (the error bars in the y-direction), and the slope of the linear regression is only 0.45. The smaller standard deviations in the model represent weaker fluctuations over the daily mean. This will be discussed further in Sect. 4.2.3 when looking into the model-measurement comparison at higher temporal resolutions.

Figure 5(c) & (d) show that the daily mean modelled values of $XCH_{4,\text{sla}}^{S}$ (solid lines in Fig. 5(c)) agree well with the daily mean measurements. The model is able to capture most of the variations in the daily mean values, while in general the observed values are slightly higher, with a linear regression slope of 0.73 and a negative MB (-1.8 $\pm$ 4.0 ppb). This small bias could be caused by the initial and lateral boundary conditions from CAMS, or due to unknown or underestimated emissions. Comparing $CH_4$ in the CAMS product with in-situ observations in the troposphere, Gałkowski et al. (2021) also reported a negligible MB, but a relatively large standard deviation (0 $\pm$ 14 ppb) in their setup.

### 4.2.3 Comparison between model and measurements: intra-day concentrations

In order to obtain a more detailed view on how the model behaves at higher temporal resolution, the longest stretch of consecutive measurements in the campaign (i.e., from 16 to 22 August) are analyzed here. Figures 6 & 7 show the daily curves of $XCO_{2,\text{sla}}^{S}$ and $XCH_{4,\text{sla}}^{S}$ at five sites for these 7 consecutive days against the corresponding modeled values with hourly temporal resolution. Model-measurement comparisons for the rest of the days are shown in Figs. S5 & S6 of the supplement.

Due to the restriction of SZAs and the corresponding availability of measured values provided by the MUCCnet (cf. Sect. 4.1), these model-measurement comparisons of the total column concentrations for GHGs can only be made during the daytime, approx. from 6:00 UTC to 17:00 UTC (cf. Fig. 6). As mentioned in Sect. 4.2.1, the modelled and observed slant column concentrations ($XCO_{2,\text{sla}}^{S}$ and $XCH_{4,\text{sla}}^{S}$) used for the model-measurement comparisons are smoothed using the SZA-dependent AK based on Eq. 3. Figures 6 & 7 also illustrate the contributions to the total column concentrations of $CO_2$ & $CH_4$ ($XCO_2$ & $XCH_4$) from different tracers in the model throughout day and night, which are calculated based on Eq. 1. This can be used to interpret the model and the measurements, e.g., the contribution of nighttime vegetation respiration to the changes in total column concentrations of $CO_2$. As described in Sect. 4.2.2, a MB of 3.7 ppm in $CO_2$ has been found over all the available measurement dates (see Fig. 5), which is defined to be the difference between the measured daily mean $XCO_2$ and the modelled values. To eliminate the bias (too high modelled background $CO_2$) and focus on the model-measurement differences due to other causes, this MB is subtracted from the modelled $XCO_2$ in the day-by-day model-measurement comparison for all sites and for each simulation date.

Figure 6 shows the modelled $CO_2$ column concentrations from all tracers. In general, there is little difference in the column background concentrations among the five sites over these six continuous days (black lines) with an MB and its standard deviation of 404.8 $\pm$ 0.19 ppm. Variations of the modelled total $CO_2$ corrected by the MB (solid red lines) are mostly dominated by biogenic activity (dashed blue lines), with only a minor influence predicted from anthropogenic emissions (dashed yellow lines).

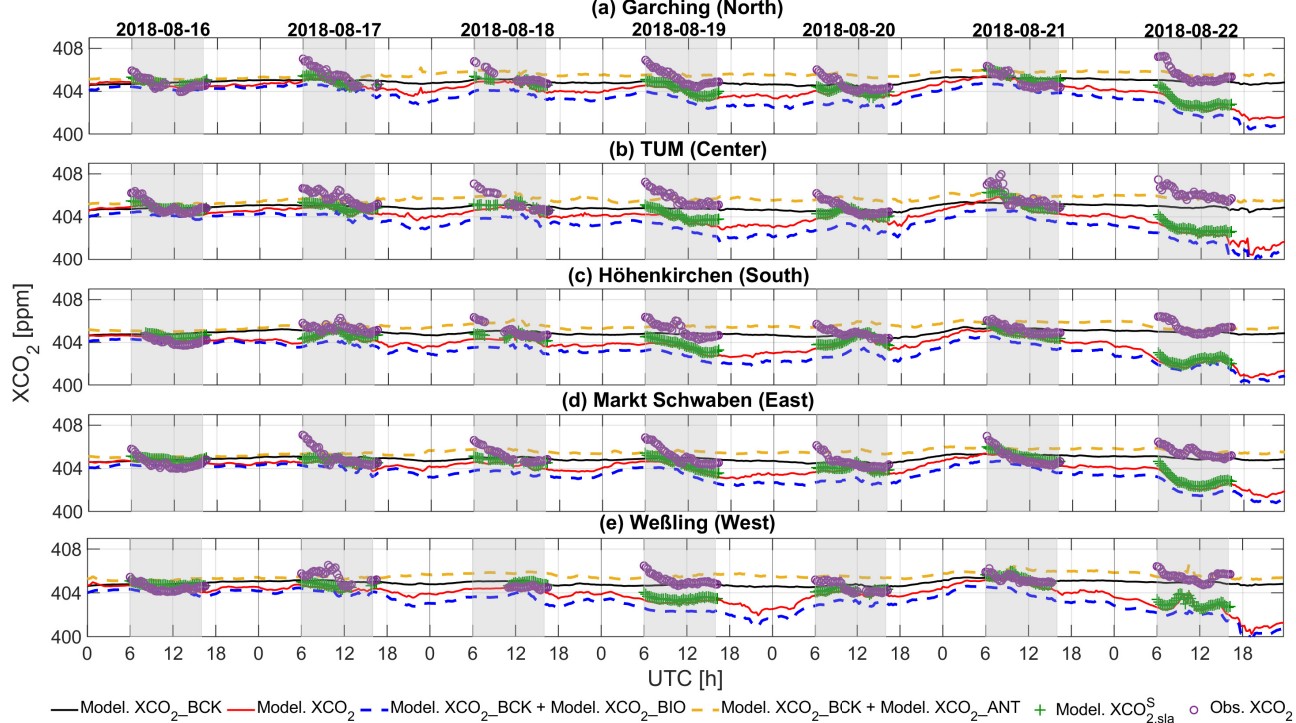

**Figure 6.** Modelled $XCO_{2,sla}^{S}$, attribution of variations to different tracers, and the measurements at the five MUCCnet sites from 16 to 22 August, 2018. The purple circles represent the column measurements from MUCCnet and the green '+' stands for the modelled $XCO_{2,sla}^{S}$ after subtracting the MB. The other lines in the plot give pressure-weighted modelled column concentrations along the full time series. These are calculated following Eq. 1 (i.e., without smoothing using the SZA-dependent AK) and are corrected by MB. The black curve represents the modelled background ($XCO_2\_BCK$), and the red curve shows the modelled $XCO_2$. The dashed yellow and blue curves highlight the concentration changes caused by human activities ($XCO_2\_ANT$) and biogenic activities ($XCO_2\_BIO$). The grey shaded areas mark the measurement periods used for comparing observations to model results.

After smoothing, the modelled, bias-corrected $XCO_{2,sla}^{S}$ (green '+') is slightly higher than the corresponding $XCO_2$ mod-
355 elled values (solid red lines), with a RMSE of 0.37 ppm and a MB with its standard deviation of $0.34 \pm 0.13$ ppm for these 7 consecutive days. This is caused by the less steep shape of the vertical profile of the AK under larger SZAs. There is no obvious difference between the modelled values with and without smoothing during the daytime from around 6:00 UTC to 17:00 UTC, which covers most of the period of the day during which measurements can be made.

The modelled $XCO_{2,sla}^{S}$ (green '+') reproduces the variability in the measurements (purple 'o') reasonably, with a RMSE of
360 1.33 ppm, a MB and its std of $-0.79 \pm 0.14$ ppm, and a coefficient of determination ($R^2$) of 0.43, as it turns out. However, the measurements often show a steep decrease in concentration during the morning, while the model only shows slight declines. This difference could be the result of an underestimation of the modelled biogenic respiration (RES) from VPRM. During the growing season (June-September), VPRM (Mahadevan et al., 2008) has been found to underestimate RES, especially at

nighttime and overestimate GPP during the daytime (Hu et al., 2021). Gourdji et al. (2021) found the differences of RES at nighttime in summertime between the improved and traditional (use in this study) VPRMs can reach more than 3 $\mu$mol/(m$^2$·s), depending on vegetation types. This causes an overestimation of the magnitude of NEE (i.e., the difference between RES and GPP), which could explain the difference between the modelled column concentrations and what is observed by MUCCnet. In our case, the observations suggested higher RES fluxes at nighttime than what was simulated by the model. This led to much lower modelled column concentrations in the early morning, which was also seen in Hu et al. (2021).

Compared to the modelled values for the other six dates, a slight rise was seen (approximately 1.5 ppm on average) in the morning of 21 August. This was induced by a combined effect of elevated background and biogenic tracer mole fractions. Closer analysis of this case (see the animation of XCO$_2$_BCK from 19 to 21 August in the supplement) has shown that the background enhancement entered from the northwest of the outermost boundary in the morning of 19 August and was transported into the innermost domain by the late afternoon of 20 August. This enhanced background signal contributes around two thirds of the modelled 1.5-ppm rise. The rest is the result of air masses with a strong biospheric CO$_2$ signature coming from the northwest of our outermost domain. Animations of XCO$_2$_BIO and XCO$_2$_BCK from 19 to 21 August are attached in the supplement to further illustrate this as well.

On 22 August, compared to the simulations for the other days, the modelled $\text{XCO}_{2,\text{sla}}^{\text{S}}$ shows a larger deficit with respect to the measurement in Fig. 6. The model can capture the variation during the day quite well, but produces too low XCO$_2$ values when bias corrected through subtraction of the MB. This more extreme mismatch is probably due to the advection of air masses heavily impacted by biogenic activities (and thus with less CO$_2$) in the model, coming into the domain from e.g., Italy, Slovenia and Croatia. As can be seen in the map of the modelled NEE from VPRM in Fig. S7 of the supplement, much stronger biogenic fluxes are found in the south of the outermost domain, compared to the other areas. From the view of the model, more CO$_2$ was taken up and the affected air masses (i.e., with comparatively less CO$_2$) are able to reach Munich when the wind is strong enough to drive them past the Alps. In addition, a constant MB over all days, as was applied for the model-measurement comparison, may not always be realistic. An evaluation of the signals within MUCCnet using the DCM is still possible, however, as will be discussed in Sect. 5.

In terms of CH$_4$, we conduct the same visual analysis (cf. Fig. 7). The variations of the modelled total XCH$_4$ (solid red) are dominated by anthropogenic activities (XCH$_4$_ANT, dashed yellow) and these two variations mostly overlap, since no significant signal induced by wetland emissions (XCH$_4$_BIO, dashed blue) is predicted by the model. The time series of the modelled $\text{XCH}_{4,\text{sla}}^{\text{S}}$ (green '+') shows general agreement with the measurements (purple 'o'). The measurements seem to capture stronger emission signals (e.g., on 22 August), perhaps due to gaps in our knowledge of the spatio-temporal distribution of CH$_4$ emissions. The modelled values show little diurnal variability at all sites compared to the measurements (RMSE: 6.7 ppb, MB $\pm$ std: -3.3 $\pm$ 5.9 ppb, and R$^2$: 0.31). Regarding the strong enhancements observed by the instruments during the daytime, especially on 22 August, these might be the result of sources which are missing from inventories, or are underestimated in their magnitude. In Sect. 5, we attempt to detect such unknown or underestimated emission sources over the domain using DCM.

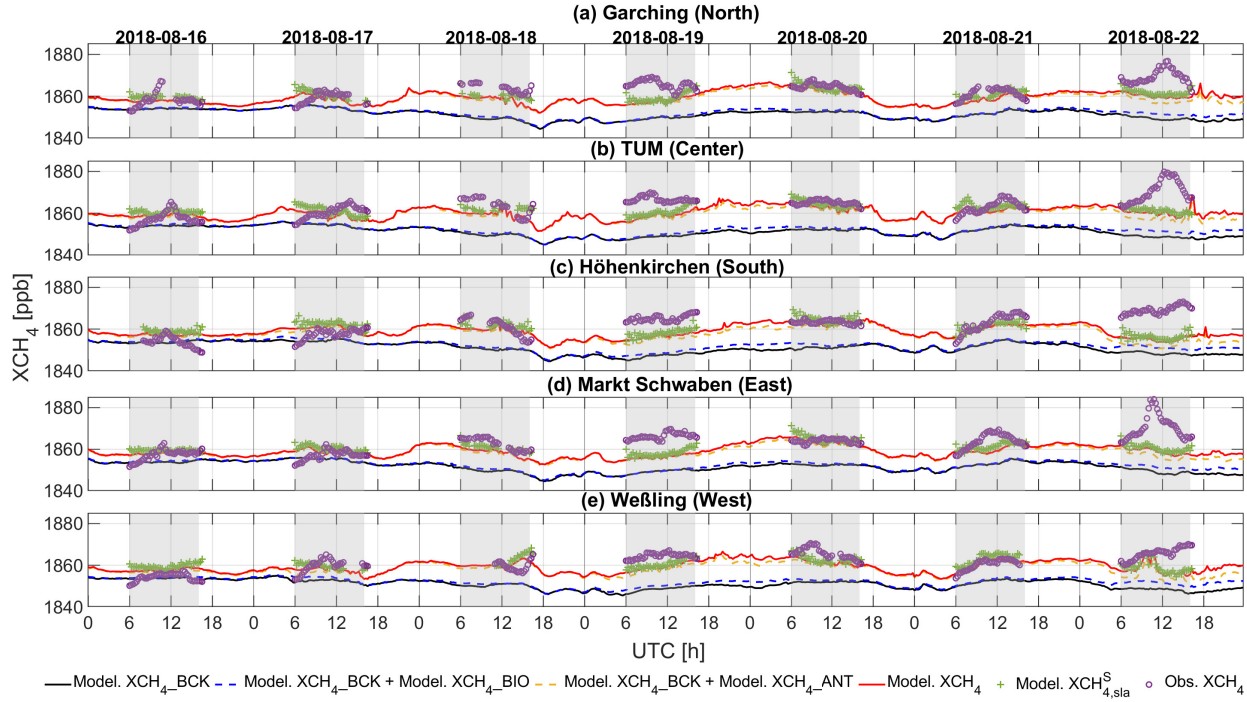

**Figure 7.** Modelled $XCH_{4,sla}^S$, attribution of variations to different tracers and the measurements at five sites of the MUCCnet from 16 to 22 August, 2018. The purple 'o' represents the column measurements from MUCCnet and the green '+' stands for the modelled $XCH_{4,sla}^S$. The other curves in the plot show the modelled column concentrations calculated following Eq. 1, i.e., without smoothing using the SZA-dependent AK: black curve – modelled background ($XCH_4$_BCK); red curve – modelled $XCH_4$. The dashed yellow and blue curves highlight the concentration changes caused by human activities ($XCH_4$_ANT) and induced by biogenic activities ($XCH_4$_BIO), including the background. The grey shaded areas mark the measurement periods used for comparing to the simulations.

## 4.3 Tracer analysis related to human activities

Beyond the major contributors to the concentration enhancements above the background as discussed above, we also analyse
the contributions from individual anthropogenic emission processes to understand how these processes impact concentrations quantitatively. To be specific, we use the GNFR emission categories from TNO-GHGco for separately advected tagged tracers. For $CO_2$, the categories are: "A. Power Plants", "B. Industry", "C. Other Stationary Combustion", "D. Road Transport" and "E. Other". For $CH_4$, the emission processes are: "A. Power Plants", "B. Industry", "C. Agriculture", "D. Waste Management", "E. Fugitives and solvents", and "F. Other".

The changes in concentrations induced by different human activities along the full time series are plotted in Fig. 8. For $CO_2$, the emissions from road transport (red) contribute the largest portion (around 37.4 %) of the total concentration enhancements caused by anthropogenic activities in August, 2018. This is consistent with the finding that over 30 % of the total GHG

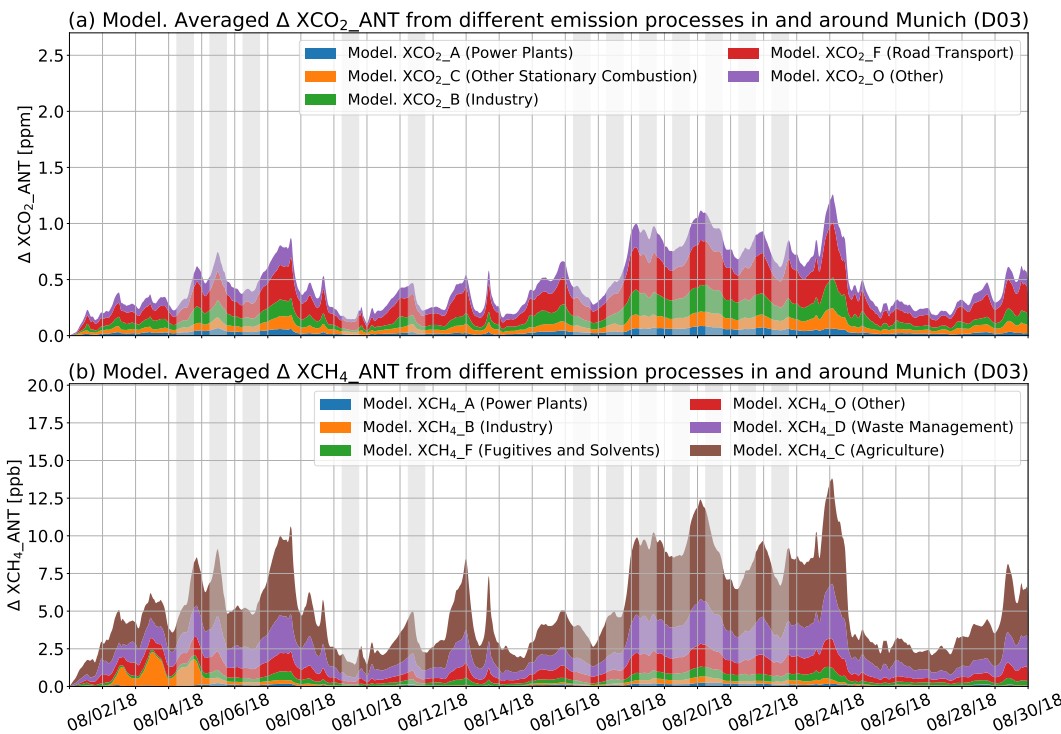

**Figure 8.** Average modelled concentration contributions by individual anthropogenic emission processes of (a) $CO_2$ ($XCO_2\_ANT$) and (b) $CH_4$ ($XCH_4\_ANT$) from 1 to 30 August, 2018 in and around Munich (D03). The grey shaded areas mark the measurement periods used for comparing to the simulations.

emissions are induced by on-road transportation for around one-third of 167 cities worldwide (Wei et al., 2021). This finding may also be due to the seasonal features of emissions in Munich, as there is no heating and less electricity generation in summer.

For other contributors, power plants account for around 7 %, other stationary combustion for 13.1 %, and both industry and other contributors for approx. 21 %, respectively. Figure 8(b) shows that for $CH_4$ the emissions are dominated by agriculture (brown) and waste management (purple), which are estimated to contribute approx. 50 % and 25 % of the total human-related concentration enhancements.

## 5   DCM-based evaluation of models, detection and tracing of additional emission sources

In order to analyse the differences between the measurements and the model, described in Sect. 4.2, we apply DCM. This method allows for a clean interpretation, eliminating biases such as the offset discussed for $CO_2$ in Sect. 4.2.2. Combining this approach with particle transport modelling using STILT (Fasoli et al., 2018),we explore how we can trace unexpectedly high measured $CH_4$ signals (cf. Sect. 4.2.3, Fig. 7) to potential additional sources that are not included in the emission inventory.

In general, DCM is an approach used to assess the emissions over a certain area through the concentration differences (gradients) between down- and upwind GHG measurement sites (Chen et al., 2016; Dietrich et al., 2021). In Zhao et al. (2019), DCM was shown to be a useful post-processing approach in model analysis and model-measurement comparisons, due to its ability to cancel out biases related to initial and boundary conditions. The aim here is to compare our measurements to the model using DCM for selected days from 16 to 22 August, 2018, before we attempt to track $CH_4$ emission sources based on this analysis with the help of STILT.

To begin with, we select up- and downwind sites for our analysis based on the surface wind data presented in Fig. 3 (Sect. 5.1). Furthermore, we select the applicable dates by using the transport model STILT (Sect. 5.2), driven by wind information along the whole column (Sect. 5.2.2). This accounts for wind shear. After comparing the modelled concentration gradients to the observations (Sect. 5.3), we further track the potential location(s) of unknown or underestimated $CH_4$ sources by using STILT footprint contours. These footprints mark the area of upstream fluxes that influence air masses which arrive at the measurement sites (Sect. 5.4).

## 5.1 Selection of up- and downwind sites

In DCM, we start by dividing our measurement sites into down- and upwind sites day by day, based on measured wind directions.

As seen in Fig. 5, the measured concentration values at the center site of MUCCnet (TUM) are found to always be higher than the values observed at the other sites, owing to the dense distribution of emission sources close to the city center, combined with the higher sensitivity to fluxes in the near-field of the observation location. To better understand the concentration gradients between the down- and upwind sites, we choose to exclude the center site when calculating the gradients.

The four remaining sites are grouped according to the wind directions observed at LMU. As shown in the wind rose of the measurements (see Fig. 3(a.3)), the prevailing wind directions during the daytime for our simulation period (WD30) are usually northeasterly or northwesterly. Table 1 shows the list of the down- and upwind sites for different prevailing wind conditions, which are used to calculate the concentration gradients.

Doing this, we assume that the surface winds measured in the city center are representative of the regional wind conditions over our domain during the day, and that they are sufficiently stationary for application of DCM. When rapid, regional-scale horizontal and vertical wind shifts occur, as during a summer cold front passage around the Alps, these assumptions might fail.

Therefore, we set up (Sect. 5.2) and use (Sect. 5.2.2) the STILT model (Fasoli et al., 2018) to assess transport patterns. For our discussion of $CO_2$ and $CH_4$ measurements vs. simulations, and most importantly for our attempt to locate unknown or underestimated $CH_4$ emissions (Sect. 5.4), we have thus been able to select days where the assumptions for DCM are likely to be met. In particular, our method ensures the validity of DCM for the interpretation of the $CH_4$ concentration peaks.

## 5.2 Selection of the applicable dates

Before applying DCM to any of the days from 16 to 22 August, we assessed its applicability for each day (see Sect. 5.2.2) by tracking the origin of air masses at different measurement sites with the transport model STILT (cf. Fasoli et al. (2018)), and

**Table 1.** Table of up- and downwind sites depending on wind directions

| Wind Direction | Upwind Sites | Downwind Sites |
|---|---|---|
| Northeasterly/Easterly (NE/N) | Markt Schwaben (East) Garching (North) | Weßling (West) |
| Northwesterly/Westerly (NW/W) | Weßling (West) | Garching (North) Markt Schwaben (East) Höhenkirchen (South) |

by assessing the modelled vertical wind profiles used for generating footprints (see Sect. 3.2). Here, we briefly describe the STILT setup.

### 5.2.1 STILT model setup

In our study, STILT with R code base (Version 2, as available via https://uataq.github.io/stilt/index.html#/, last access: 11, Jan 2022 ) was implemented using around 168 core hours provided by the high performance computer center LRZ. To assure transport consistency with previously presented results, STILT was driven by the WRF meteorological fields generated for our second domain (D02) at a horizontal resolution of 2 km. An extended discussion of WRF-Chem and STILT is included in Sect. S11 of the supplement. In order to trace back the origin of air masses at a given spatio-temporal receptor point (corresponding to the time at which an instrument performs a given measurement), STILT uses ensembles of tracer particles which are propagated backwards in time. Specifically, the model provides us with the sensitivity of the analysed slant columns to emissions in lower part of the planetary boundary layer height. These sensitivities are calculated by considering the residence time of released particles when they traverse the lower planetary boundary layer before reaching the measurement location, and are further aggregated over the STILT simulation time to produce footprint maps. In our configuration of STILT, we have released 500 particles at 13 altitudes along the slant column, namely at 20, 180, 350, 520, 700, 880, 1060, 1250, 1440, 1620, 1920, 2020 and 2220 m above the ground level for each simulation.

STILT then yields so-called footprint contours (i.e., contours enclosing a certain percentage, e.g., 90 %, of the accumulated surface sensitivity) for each altitude layer. Details of the percentile footprint contours are described in Sect. S12 of the supplement. In order to yield an estimated effective footprint independent of emission height (depending on source and local conditions), we have aggregated these altitude-dependent footprint data for the different layers using the pressure differences between layers as weights (Jones et al., 2021).

### 5.2.2 Date selection with footprints from STILT

Leveraging the footprint contours from STILT, the differences in the origin and path of air masses arriving at up- and downwind sites can be determined. Understanding these differences is a key prerequisite for determining the location of potential unknown or underestimated GHG sources based on noteworthy signals in the downwind-upwind concentration gradients. Whenever air

masses reaching up- and downwind sites have very different areas of influence, the upwind site cannot be used as a relative background site when calculating concentration gradients. In addition, in cases where there are emitters located upstream of an upwind site, the presence of the strong local signal prevents it from being used as a background site. When the footprints fully overlap, however, such that air passes over the upwind to the downwind site, and additional GHG contributions are from in between, DCM can be used. But even when this condition is not strictly met, if differences in the footprint areas are small, the small non-overlapping parts are potential locations for unknown or underestimated GHG emitters and sinks to be pinned down, rendering DCM effective. Signals coming from the overlapping area of the footprints, in contrast, will be visible at multiple measurement sites with a characteristic time delay. Clearly, for all this to hold, we need to check whether our footprints are realistic. The main prerequisite for this is the accuracy of the WRF wind fields, driving STILT, at different heights. We check this criterion at the end of this section using radiosonde data (see Sect. 3.2).

The approach outlined above can be applied to understand up-/downwind differences and obtain information about GHG sources and transport in the target area. Here, we adopt the following strategy: for each measurement site, we compute the footprint contours with the receptor time in STILT set to the time of the daily $XCH_4$ peak value [1]. Then, we accept only days for our study where the overlap of these footprints is large. The peak times of the stations are usually different by only a few hours.

Our strategy results in three outcomes: $(i)$ days with unstable wind conditions in time or with large variations of wind directions from one site to another are excluded; $(ii)$ the large overlap and small differences in footprint contours allow for a clear localisation of potential origins of differences, as discussed above; and $(iii)$ we can understand whether the peak is of the same origin at all sites. We chose to analyse $CH_4$ signals at the end of this study (Sect. 5.4), as this offers a realistic possibility of tracking human emission sources (cf. Sect. 4.2.3) in this exploratory work. In contrast, the current state of the art makes it more difficult to trace anthropogenic effects in $CO_2$ signals, where biogenic activity plays a much larger role.

We applied our strategy to all measurement days, and finally selected 16, 20 and 21 August 2018 as days suitable for further analysis (Sects. 5.3 and 5.4). Figure 9 and the supplement (in particular Fig. S13 and Sect. S12) lay out the reasons for our decision and show all the footprint contours. In the following paragraphs, we discuss the examples from Fig. 9 (16 and 22 August) further as typical days deemed appropriate (16 August) or inappropriate (22 August) for further analysis using DCM. Besides the footprints, figure 9 also shows the peak times used as receptor times for STILT.

On August 16, with easterly wind prevailing, the instruments deployed upwind (Garching/Markt Schwaben) captured peak $CH_4$ signals in the first half of the day and a similar signal was then seen at TUM about an hour later (red dots in Fig. 9(a)). However, the sensor at the downwind site (Weßling) did not detect a major peak. Using the knowledge of station locations and the observed peak times, we would have predicted a peak at Weßling (Fig. 9(a), black dot) as follows: the upwind and central sites captured the peaks at around 11:00 and 12:00 UTC, respectively. These air masses would then possibly have reached the downwind site after approximately two hours (estimated from wind speeds as given in the supplement, Table S4),

---

[1]In a few cases where the peak times have been hardly detectable, they have been inferred using the peak times at nearby sites and considering time delays derived from the daily surface wind speeds from LMU.

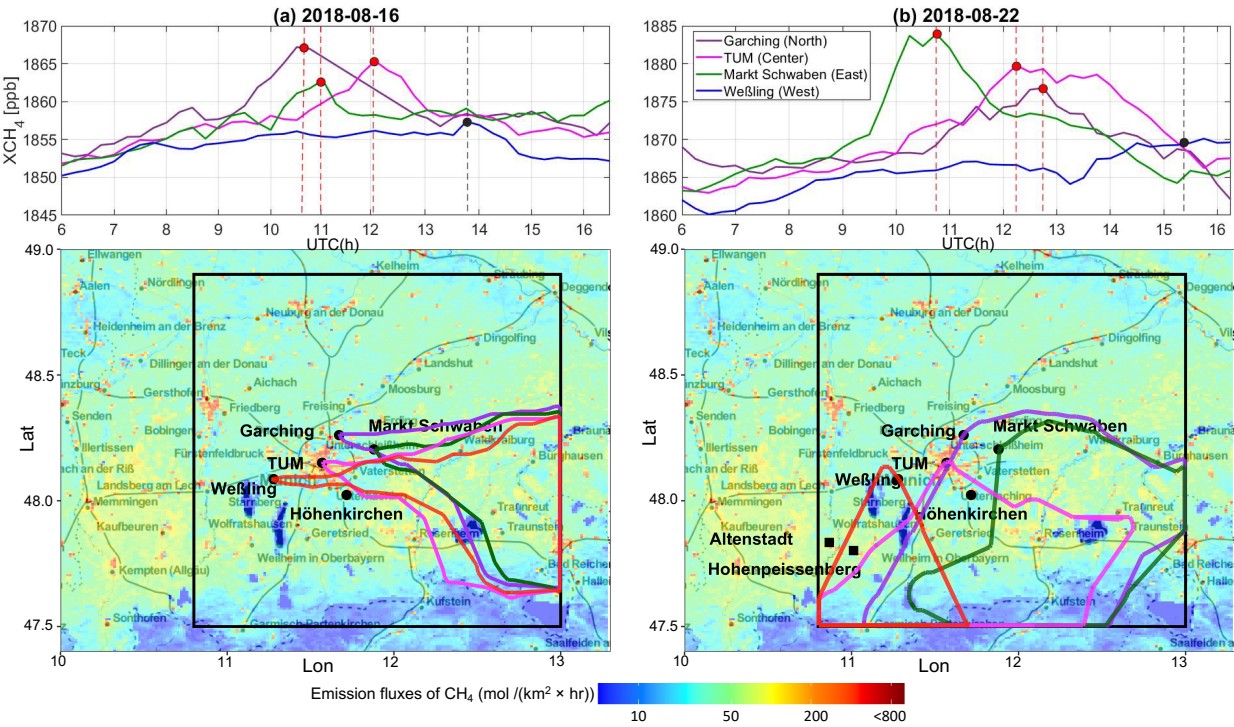

**Figure 9.** Observed XCH$_4$ over time (upper panel) and the 90th percentile contours of column footprints (lower panel) on (a) 16 and (b) 22 August at up- and downwind sites with different colors: red for Weßling (West), green for Markt Schwaben (East), purple for Garching (North) and pink for TUM (Center). The peaks in the observations are marked by dots (red: clear peaks, black: inferred peaks, see main content/footnote). Two black squares in (b) mark the measurement sites of IGRA (cf. Sect. 3.2). The background maps use tiles from Stamen Design (https://maps.stamen.com/, under CC BY 3.0, with data by OpenStreetMap, under ODbL, from 2021 Dec. 21). The map colouring reflects the emissions from the initial inventory (i.e., TNO-MACCco). The STILT model domains are marked by the thick black boxes.

corresponding to the distance between TUM and Weßling, i.e., at 14:00 UTC (Fig. 9(a), black dot). Note that the receptor times for all up- and downwind sites and dates, determined from observed peaks or estimates in this manner, are listed in Table S7.

On 22 August, the origins of air masses at most of our sites differ significantly (Fig. 9(b)). The footprints (with receptor times corresponding to the red dots in Fig. 9(b), top panel) do not overlap as cleanly, and any 'tracing experiment' would be poorly controlled. The peaks seen at the Markt Schwaben (East), TUM (Center), and Garching (North) sites may well be of different origin, given the different footprint shapes. Thus, we do not study this day in further detail.

### 5.3   Model-measurement comparison of concentration gradients

After checking the prerequisites in the previous sections, we focus our analysis of the differential concentrations (gradients) for the selected dates of 16, 20 and 21 August.

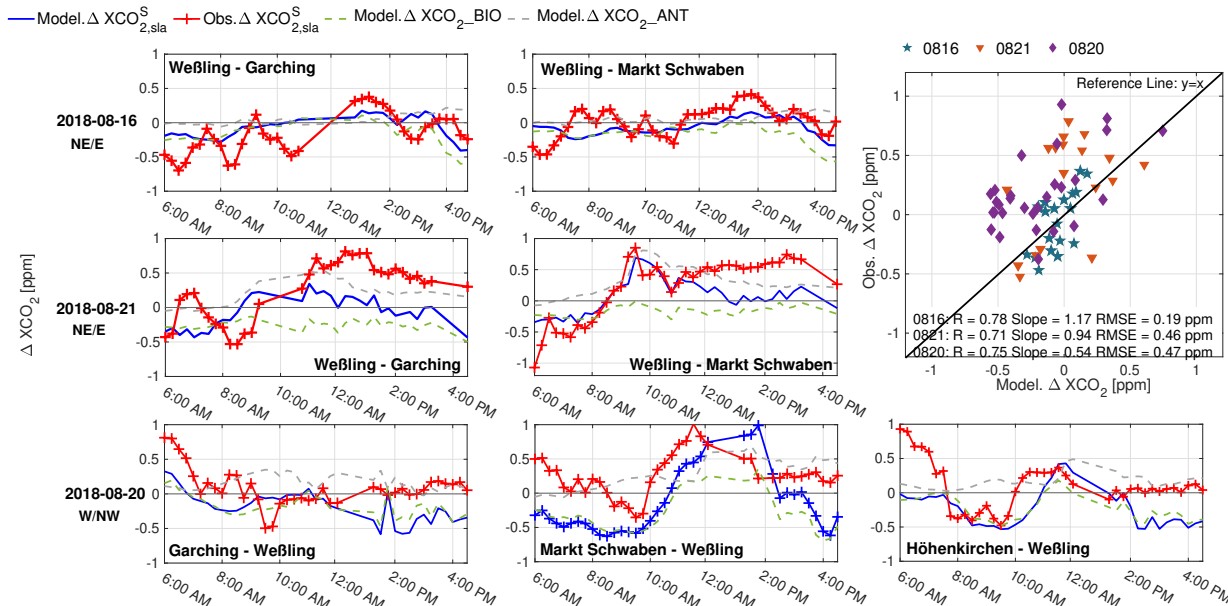

**Figure 10.** Time series of $\Delta\mathrm{XCO}_{2,\mathrm{sla}}^{\mathrm{S}}$ for three targeted days and their scatter plot: 16 and 21 August with NE/E winds in the upper two rows, and 20 August with W/NW winds in the bottom row. The column concentration differences between the down- and the upwind sites are plotted as red solid lines for measurements, blue solid lines for the modelled full signal values, light green dashed lines for the simulated biogenic signal values and grey for the modelled contributions related to anthropogenic activities.

Figure 10 shows $\Delta\mathrm{XCO}_{2,\mathrm{sla}}^{\mathrm{S}}$ for the days selected based on the prevailing wind directions. The modelled concentration gradients of $\mathrm{XCO}_{2,\mathrm{sla}}^{\mathrm{S}}$ between the down- and the upwind sites (blue solid lines in Fig. 10) are driven by both biogenic activities (light green) and human activities (grey). The biogenic part can be attributed to the special spatial distribution of biogenic sinks
in Munich and its surroundings (see also Fig. S6 of the supplement): The southeastern and southwestern parts around Munich are more biologically active and have greater carbon sinks, compared to the other areas. This is an interesting difference to Berlin, where our previous study (Zhao et al., 2019) showed no such signal, corresponding to a relatively even distribution of biogenic fluxes over Berlin. In the model-measurement comparison of $\Delta\mathrm{XCO}_{2,\mathrm{sla}}^{\mathrm{S}}$, the model (blue) was able to reproduce the general variations when comparing to the measurements (red) shown in Fig. 10, with a Pearson correlation coefficient (r) of
0.74 and RMSE of 0.37 ppm. However, the modelled and measured concentration gradients show some differences, e.g., the modelled concentration gradients between Höhenkirchen and Weßling before 10:00 UTC on 22 August were underestimated compared to the observations. The differences between the modelling results and the measurements could be potentially caused by underestimated concentration gradients from biogenic fluxes in and around the city (light green) in the model. In particular, underestimating RES during nighttime (details in Sect. 4.2.3) could result in the underestimation of concentration gradients of
$CO_2$ in the early morning.

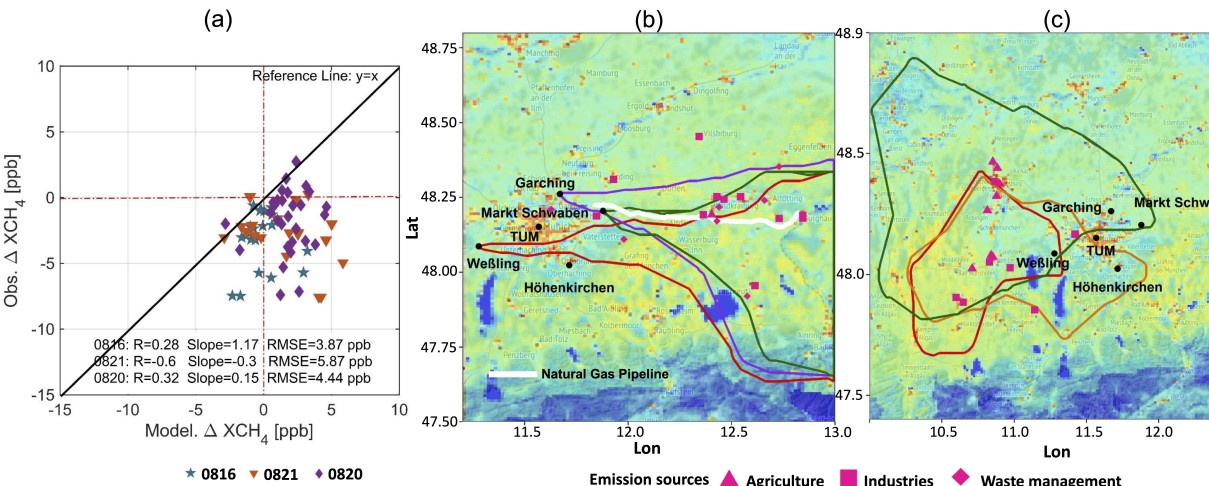

**Figure 11.** (a) Scatter plot of modelled and measured $\Delta\text{XCH}_{4,\text{sla}}^{\text{S}}$ for three targeted days and distribution of known emission sources located in the (b) eastern and (c) western area of Munich and the footprint contours for down- and upwind sites on (b) 16 and (c) 20 August: green for Markt Schwaben (East), red for Weßling (West), purple for Garching (North) and yellow for Höhenkirchen (South). The white solid line in (b) shows the location of the Burghausen-Finsing-Amerdingen high pressure natural gas pipeline. The background maps use tiles from Stamen Design (https://maps.stamen.com/, under CC BY 3.0, with data by OpenStreetMap, under ODbL, from 2021 Dec. 21). The map colouring reflects the emissions from the initial inventory (i.e., TNO-MACCco), with the same color bar as in Fig. 9.

As seen in the scatter plot of $\Delta\text{XCH}_{4,\text{sla}}^{\text{S}}$ in Fig. 11(a), the modelled values are mostly positive during the day, while the measured concentration gradients between the down- and the upwind sites are mainly negative. The times series of modelled and measured $\Delta\text{XCH}_{4,\text{sla}}^{\text{S}}$ are shown in Fig. S14 (see Sect. S13 of the supplement). That is, the instruments always measure strong signals at the upwind sites compared to the downwind sites, which cannot be reproduced by the model. As a large

methane sink over the city is not expected, the most likely cause for this phenomenon is that emission sources located upstream of an upwind site (i.e., somewhere to the northeast or east of the Garching and Markt Schwaben stations in the case with NE/E winds) are missing or underestimated in the initial emission inventory. Likewise, with W/NW winds, the negative measured concentration gradients between the three down- and one upwind sites are found with -1.89 ppb in daily means and the model fails to reproduce these signals. Again, the measured column concentrations at the upwind sites (i.e., Weßling) are generally

higher than at the downwind sites. Especially in the morning of 20 August, a clear strong increase was captured at the upwind side (see Fig. 7). However, none of these features could be replicated by the model. We postulate the presence of an unknown or underestimated source of emissions located upstream of Weßling as the most likely explanation.

In addition to the errors caused by the uncertainties in the initial emission inventory, other potential causes could contribute to errors in the concentration and thus, the gradients as well. The bias brought by the modelled meteorological fields can

contribute to the error of the modelled column concentrations and further to discrepancies in the gradients (Wu et al., 2018). Further, our current DCM approach does not take the transport time into account. Moreover, factors such as the mixed layer height and topography could also introduce biases (Hedelius et al., 2017).

## 5.4 Localizing unknown/underestimated emission sources

To further localize the underestimated or unknown emissions at the upstream areas of both upwind sides (i.e., the western area of Weßling and the eastern area of Markt Schwaben), the footprint contours are used to interpret the transport of air masses. Figure 11 shows the footprint contours of the up- and the downwind sites with two different wind conditions (a) for 16 and (b) 20 August and their receptor times are listed in Table S7 of the supplement.

In this study, the modelled contributions from human activities are initialized with the emission fluxes from the emission inventory TNO_GHGco_v1.1 for the year 2015. The multinational spatially, temporally explicit emission inventory holds large uncertainties, due to the large variability in spatiotemporal distributions of $CH_4$ emissions from different sectors in different regions that have not yet been fully captured by the emission inventory (Bergamaschi et al., 2022), the disaggregation from annual emissions to hourly values by using temporal profiles, and the temporal inconsistency of emission information from 2015 or even earlier than the study period in 2018. This could results in missing or underestimated emissions, as suggested by the measurements.

After delineating the areas where the uncertain sources could be located, they were further pinpointed based on the updated database and local knowledge. As mentioned in Sect. 4.3, the major contributors of $CH_4$ related to human activities are waste management, agriculture and industries (USEPA, 2019). A number of sources from these sectors in and around the areas covered by the footprint contours are identified based on the European industrial emissions portal (available as https://industry.eea.europa.eu/, last access: 14 December, 2021), the initial emission inventory and local knowledge, which are marked in Fig. 11(b) & (c). The waste management here refers to landfill and wastewater treatment facilities, the emissions induced by agriculture come from livestock and the industry emissions are from heat and gas production, manufacturing, etc. Another potential source of the observed signals that was not represented in our emission inventory was a high-pressure natural gas pipeline that passes through the eastern section of our domain. This pipeline was reportedly under construction and tested in 2018 (constructed by Bayernets GmbH; Macht (2017); Bayernets (2018)).

Clearly, as a preliminary study, we propose the application of this approach in brief, but cannot yet determine the exact cause of the observed peaks that were not reproduced in the model. However, with a longer observation record and refinements to the modelling approach, we see the potential to track down strong emitters of GHGs. It should be noted, however, that accurate estimation of unknown or underestimated sources needs to be performed using a combination of observations and a quantitative footprint analysis. This could provide information for supporting mitigation strategies. Here, the year-round measurements from MUCCnet, which cover a wide range of weather conditions and complete seasonal cycles, will help to complement and improve general inventories.

## 6 Conclusion

We have developed a WRF modelling framework for Munich to accompany MUCCnet and provide regularly updated concentration maps. Compared to a previous study for the city of Berlin (Zhao et al., 2019), we have introduced additional tagged tracers and improved model inputs. Measurements from MUCCnet and meteorological stations have been used to validate

the model. Simulated slant column concentrations, extracted from the model with a smoothing consistent with the instrument characteristics, show encouraging general agreement with observations. We have then focused on comparing modelled column concentrations to measurements and identified the flux categories responsible for the observed signals. The diurnal cycle of $CO_2$ was not well captured, with the model showing lower concentration enhancements in the morning, while agreeing well in the afternoon. We suspect that this is related to underestimated nighttime RES fluxes in VPRM, which has been reported in another recent study.

Our study concludes with a refined application of DCM, aided by air-mass transport tracing with the STILT model. Despite the continuous total-column measurements surrounding the city center, this analysis highlighted the challenge of extracting the anthropogenic signal from such data. Even though our model was not able to fully reproduce the measured gradients in $CH_4$ over this complicated source region, this exploratory application enabled us to identify unexpected signals in the measurements and to roughly delineate the potential uncertain source regions in the inventory. The outcomes of this study may provide guidance for other groups considering the optimal instrumentation and analysis frameworks for measuring urban anthropogenic signals.

This study focused on a one-month period of measurements to refine the modelling approach, but the continuous measurements of MUCCnet are ongoing. This increasingly long and complete data set will enable emission sources to be monitored on a regular basis. The evaluation of model output and measurements for longer time periods, with various wind conditions, will allow for a better localisation of sources, and for an improvement of emission inventories. We are looking forward to conducting further studies in this direction, in particular as Munich is one of the three main pilot cities in the EU H2020 project ICOS Cities. Within this project, more measurement sites will be operated in and around Munich in the coming years. More comprehensive comparisons for meteorological parameters can be expected. A study into the use of the simultaneously measured total column carbon monoxide (XCO) to constrain emissions from combustion processes can be carried out. Inclusion of measurements from aircraft (e.g., In-service Aircraft for a Global Observing System (IAGOS) project – www.iagos.org) and satellites (e.g., the Orbiting Carbon Observatory-2/3 (OCO-2/3) operated by NASA) will help to further validate and make use of our models. Our data-model framework will bring us closer to the aim of effectively tracing GHG emissions and improving emission inventories.

*Author contributions.*

The modelling framework designed for Munich is mainly built by XZ with joint effort from JM, MG, CG, SH and JC. Regarding the initialization, JM, MG and CG retrieved the CAMS fields, and preprocessed the vegetation indices and the necessary variables for the Kaplan model. JC, DF and AW helped with the interpretation of the measurements. CG, JG, JM, AS and XZ put the effort in the improvement of VPRM. SH gave guidance on running the WRF model on the Linux Cluster of Leibniz Supercomputing Centre (LRZ, Garching b. M., Germany). XZ, JC and SH designed the computational framework. XZ and JC performed the analysis of the results with support of JM, CG and MG. With the input from all authors, XZ wrote the manuscript. All authors provided critical feedback and helped shape the research, analysis and manuscript.

*Data availability.*

The model data used to support the results described in this paper are available upon request to the first and corresponding authors. The measured column-averaged dry-air mole fractions of $CO_2$ and $CH_4$ recorded during an urban measurement campaign in Munich in August 2018 can be found in Dietrich et al. (2023).

*Competing interests.*

The authors declare that they have no conflict of interest.

*Acknowledgements.* TUM authors as well as MUCCnet are supported by the German Research Foundation (DFG, grant nos. CH 1792/2-1, INST 95/1544).. WRF and STILT workflows were run on the CoolMUC-2 High-Performance Computing (HPC) system, part of the Linux Cluster at LRZ, where also two nodes of ESM are housed.

Ankit Shekhar acknowledges funding by ETH Zürich project FEVER ETH-27 19-1.

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
