# Peer review of "Understanding greenhouse gas (GHG) column concentrations in Munich using WRF"

_Atmospheric Chemistry and Physics, 2022_

## Author Comment (AC1)

**General comments**

This paper describes a modelling framework based around WRF that will be used with a network of spectrometers for top-down monitoring of greenhouse gases in Munich. The methodology based around WRF-CHEM and STILT with other datasets for emissions, land cover, boundary conditions etc. and data from the spectrometer network was introduced in a previous paper by the authors. The modelling framework is clearly outlined and some best practices for top-down emission monitoring are described. In particular, the authors describe some qualitative checks for determining when a gradient method may be appropriate for top-down measurements. These concerns can be of interest to other groups working on urban or regional carbon observations. The paper also makes some comparisons of their meteorological fields and modelled greenhouse gas signals to measurements from their networks on a few days in August, 2018. The paper also suggests explanations for discrepancies they observe between their measurements and model. However, some of these explanations are not strongly supported and the authors may need to consider some alternative explanations to make this aspect of the paper stronger.

Additionally, although the authors describe two models in their paper, there is no discussion contrasting the approaches or explaining the differences in results that the authors find with them. The authors also find that their gradient method is unable to isolate signals from their region of interest during their measurement days. I think that a discussion of why this is and how such a network could be improved or the challenges creating a network at a 10's of km scale in a complex source region would be very useful.

The paper is well-written without many typos and well structured for the most part.

We thank the anonymous Referee #1 for their time and valuable comments to improve this manuscript. We have improved our explanations following your kind comments and suggestions in this revision. The general and specific comments are addressed point-by-point in replies below. The referee's comments have been repeated in black. The authors' replies are marked in blue and the edited contents of the manuscript are documented in red in tables below each comment. Moreover, we set the numbers of the figures in this revision as 'R' plus the numbers (e.g., Figure. R1), while the figures in the manuscript are numbered with 'M' plus its numbers (i.e., Figure. M1).

We assume that the "two models" that the reviewer is referring to in the comment above are WRF-Chem and STILT. We have done our best to address this comments and concerns, making a table to summarize the differences of these two models (from general objectives to technical details), which we explain replying to the specific comments below (see Pages 9 & 10).

Following the valuable comments from the referee, we have extended the explanation regarding the model-measurement biases, especially the part related to the overestimation in the modelled background $CO_2$ concentrations. It is true that DCM was not able to isolate the signals from the area of interest. The question of how measurement networks similar to MUCCnet could be improved, and the challenges in creating a network at the city level in a complex urban region is beyond the scope of the current study. This is an exploratory analysis, based on a limited number of measurements (one month measurement campaign). Building on the methods outlined in this study, work is currently underway to develop more sophisticated modelling approaches, which will be the focus of future studies. Furthermore, the MUCCnet data and the related modelling systems are involved in several ongoing projects. One in particular (ICOS Cities, https://www.icos-cp.eu/projects/icos-cities) is addressing the problem of network design directly, by comparing pilot projects in various cities. The outcomes of this project may provide further guidance for optimal instrumentation and analysis approaches.

**Specific comments**

Figure M1: include a scale bar please. I also suggest you remove extraneous land use categories (eg tundra) from your legend as it is confusing and use a different colour scale (eg greens for vegetation etc.). Jet could be interpreted as a continuous scale but land cover is discrete and categorical. As the figure is drawn now, it is difficult to tell what land covers are what colour on your map. I would also indicate your meteorological reference stations (sondes and surface).

Response: We thank the referee for these suggestions on Fig. M1. As shown in Fig. R1, there are in total 9 LULC categories that are not present D03 and are struck through in red on the X-axis. We have removed these extraneous categories from the legend of Fig. M1 (see updated version in Fig. R2). Moreover, we have also defined our own discrete color bar for these LULC categories, as was suggested in the comment. Figure R1 and relevant information have been added to Sect. S3 of the supplement.

[Figure]

Figure R1: The contribution of each Copernicus land use land cover (LULC) category over Munich (D03). The categories that are crossed out on the X-axis are absent in D03.

Regarding the two surface stations used for the model-measurement comparison of meteorological fields, they have been marked as black dots and labelled. The LMU station is quite close to the center site of MUCCnet (i.e., TUM), as described in Sect. 3.1 of the manuscript, separated by approximately 600 meters. Thus, in addition to marking it in the figure, we also changed the text 'close to the center of Munich' to 'close to the center site of MUCCnet' in line 169. The other station is located at Munich airport, outside of the city administrative boundaries.

| Lines 176~177 | *The first station is located at the Meteorological Institute of the Ludwig Maximilian University of Munich (LMU; latitude: 48.15 °, longitude: 11.57 °, altitude: 561 m),*
 *close to the center site of MUCCnet.* |
| --- | --- |

The two reference stations of radiosondes from IGRA are not marked in Fig. M1, since they are not located in the innermost domain (D03, i.e., the right panel of Fig. R2). To see the locations of these two stations, we have marked them in Fig. M7(b).

Your footprints suggest that you are often sensitive to emissions outside of Munich and you refer to discrepancies

1.Evergreen Needleleaf   2.Evergreen Broadleaf   4.Deciduous Broadleaf   5.Mixed Forest   6.Closed Shrubland   7.Open Shrubland

8.Woody Savanna   10.Grassland   11.Permenant Wetland   12.Cropland   13.Urban & Built-up   14.Cropland Mosaics

16.Barren   17.Water Bodies   19.Mixed Tundra   21.Lake   32.Compact Mid-rise   35. Open Mid-rise

36. Open Low-rise   38.Large High-rise   39.Sparsely Built   40. Heavy Industry

Figure R2: Topography map for the entire domain area (left panel). The right panel shows the land use classification in D03, including the 16 classified IGBP-modified MODIS land-cover types (from 1 to 21, as illustrated in the color bar and labels), and 6 classified LCZ land-cover categories defined for the urban areas of D03 (numbers larger than 30, as illustrated in the color bar and labels). The five measurement sites in our MUCCnet campaign and the surface weather stations used for the model-measurement comparison of meteorological fields are marked as black dots on the right panel. The national boundaries and coastlines in the left panel are from National Center for Atmospheric Research (NCAR) Graphics Version 4.1 (© UCAR/NCAR).

in the background concentrations. Therefore a comparison between two stations in Munich to your meteorology may not be appropriate for evaluating model bias. Including more stations within your domain may be better (2 would be appropriate if you were isolating signals in Munich but your initial results suggest you aren't?). I also think that this section should be combined with your discussion of the radiosonde data and you do a more holistic comparison of the meteorology based on the two. This section may also be better places after the section introducing your study area (4.1) so the reader is situated in your field site before discussing the comparison.

Response: Thank you for this valuable comment. We wholeheartedly agree with your point, unfortunately apart from the data presented in the study, additional meteorological (especially wind observations) data were not available during the analysis period of this measurement campaign. Overall, we have evaluated the modelled meteorological fields by comparing them with observations from four stations, two surface measurement stations and two radiosonde stations from IGRA that provide measurements along a vertical profile. We believe that these comparisons form a relatively complete analytical assessment of the meteorological data. More stations, planned within the "ICOS-Cities" project, will become operational in the coming years, allowing for a more robust evaluations in the future. This information has been added in the outlook (Sect. 6: Conclusion) of the edited manuscript.

| | |
|---|---|
| Lines 592~595 | *We are looking forward to conducting further studies in this direction, in particular as Munich one of the three pilot cities in the EU Horizon 2020 project ICOS Cities. Within this project, more measurement sites will be operated in and around Munich in the coming years. More comprehensive comparisons for meteorological parameters can be expected.* |

As per the reviewer's suggestion, we have moved the model-measurement comparison of the two radiosondes to Sect. 3 of the manuscript ('Model-measurement comparison for wind fields'). The details can be found in Pages 8-9 of the manuscript.

Line 181: Since your measurements are not available on every day in August, it might be appropriate to also report the biases on the days when you have measurements as well. Otherwise errors in the wind specific to the measurement days and relevant to your modelling in this paper may be masked.

Response: Thank you for your suggestion. The purpose of Fig. M2 is to assess the overall quality of the wind fields reproduced by WRF, and therefore we showed simulated data for the whole period of our measurement campaign. We did not include a screening for measurement periods, because the EM27/SUN measurements had not yet been discussed at this point in the analysis. Additionally, limiting the analysis only to the measurement days also has drawbacks, as the concentration fields in the study area are affected by circulation patterns present in the previous days as well, making it difficult to define the appropriate comparison period. However, following your remark, we marked the measurement days in Fig. M2 and re-phased the sentence about the biases on these days.

| | |
|---|---|
| Lines 195~198 | *Along the time series, the simulated (Fig. 2(b): red crosses) and measured (blue dots) wind speeds show similar variability, but the model generally overestimates wind speeds with a root mean squared error (RMSE) of 2.0 m/s and a mean absolute error (MAE) of 1.2 m/s. During the measurement periods (cf. Sect. 4) marked by the grey shaded areas in Fig. 2(b), the model performs slightly better with a RMSE of 1.6 m/s and a MAE of 1.1 m/s.* |

| | |
|---|---|
| Lines 202~203 | *Over the measurement periods marked by grey areas, smaller differences between the WRF and the in situ surface wind directions are found, with a RMSE of 58.2 ° and MAE of 1.8 °.* |

Line 235: The FTS retrieval typically uses a profile based on NCEP for the pressure and temperature profile and surface pressure from the instrument. Here you use the WRF meteorology to define the model pressure levels and surface pressure from WRF. I wonder if this discrepancy might introduce a bias in your model-measurement comparison. It may not but it is worthwhile to consider and discuss briefly.

Response: Thank you for this comment. For the five instruments deployed in MUCCnet, one pressure profile is used for the measurement retrieval. The WRF model provided pressure and temperature at a much finer spatial resolution than that of NCEP, at a horizontal resolution of 400 m and with 45 vertical levels. To investigate whether discrepancies of pressure between the values used in the retrieval and the WRF model values could have a significant impact, we extracted the pressure and temperature profiles from the ".map" file used in the retrieval Figure R3 shows the comparison between these retrieval values and the modelled values. In general, only small difference in temperature and pressure are found above around 10 km. As the modelled pressure profiles match the vertical structure of the modelled concentration fields, the modelled column concentrations can be calculated without the need for interpolation. The relevant information has been added in the manuscript as follows:

| Lines 269~273 | *Here we use WRF meteorology derived pressure profiles to calculate the weights for each vertical layer, instead of the pressure profiles used for EM27 retrievals (NCEP). This is convenient, as the WRF modelled profiles for both pressure and concentrations have the same vertical structure and no interpolation is necessary. Furthermore, only slight differences in pressure at higher altitudes was found between the WRF model and NCEP (see Sect. S10).* |
|---|---|

[Figure]

Figure R3: Comparisons of (a) Temperature and (b) Pressure Profiles between the values from NCEP and the WRF modelled values for five MUCCnet sites at 12:00 UTC of 16 August. The profiles of the MUCCnet sites coincide such that only the TUM profile (green graph) remains visible in each panel.

Line 260: Here you refer to the WACCM a-priori profile. However, in your section on the EM27, you say you use the retrieval methodology of Dietrich et al (2021). They say the use GGG2014 for their retrieval and this software typically calculates its own a-priori profiles during the retrieval. For the calculation of the averaging kernel correction, you use the a-priori that you use in your retrieval so can you clarify what you used in your retrieval in your section describing your EM27 measurement? Is it WACCM or the GGG-2014 profile?

Response: Thank you for pointing this out. For the measurement campaign in 2018, the GFIT GGG-2014 algorithm was used for the retrieval. Currently, PROFIT and GGG-2020 are applied in MUCCnet. The GFIT GGG-2014 algorithm defines its own a-priori profile, instead of using WACCM. We have clarified this information in our updated manuscript.

| Lines 233~235 | *The retrieval algorithm GFIT GGG-2014 (Wunch et al., 2015) was applied during the measurement campaign of 2018, while currently MUCCnet is using the PROFIT (Hase et al., 2004; Frey et al., 2019) and GGG-2020 algorithms (Laughner et al., 2023).* |
|---|---|

Line 286: Note that Gałkowski et al. are about in-situ concentrations. The size of the bias or variability expected for an in-situ measurement would generally be larger than a column error. Additionally, their analysis would ignore flaws in your stratospheric column so I think it is appropriate to note in your text that the comparison is not like to like.

Response: Thank you for this comment. Please note that in their study, Gałkowski et al. have reported the bias statistics for an elevated layer of the troposphere (3 km - 10 km a.m.s.l.), which based on their observations they have interpreted as being caused by discrepancies stemming primarily from offsets in far-field contributions.

Considering that a) most of the measurement sensitivity for both $XCO_2$ and $XCH_4$ is for the tropospheric part of the column, and b) we find the boundary condition from CAMS (the same product that was used in the quoted study) to be the primary source of discrepancy between our observations and the model, we believe that the very good agreement between the numbers from all reported studies (ours, Gałkowski et al. and Tu et al.) is more than a coincidence. Overall, we agree that that comparison of column variability and that of Gałkowski et al. cannot be made directly without careful consideration (in particular of the effects of errors in the stratosphere), and we clarify that in the revised manuscript, as suggested.

| Lines 310∼314 | *Gałkowski et al. (2021) found a similar bias between the CAMS product and airborne measurements in the free troposphere over Europe, with a MB of 3.7 ± 1.5 ppm, which they attributed to the far-field contributions to the local signal. While the agreement between the biases reported in their study and here is excellent, it should be noted that the numbers are not fully comparable, as the authors of the quoted study evaluated a limited section of the vertical column (namely between 3 – 10 km a.m.s.l.) using in-situ data, thus excluding the stratospheric component of the total column.* |
|---|---|

| Lines 325∼329 | *Comparing $CH_4$ in the CAMS product with in-situ observations in the troposphere, Gałkowski et al. (2021) also reported a negligible MB, but a relatively large standard deviation (0 ± 14 ppb) in their setup. As in the case of $XCO_2$, it should be noted that the comparison between these estimates should not be over-interpreted due to differences in the vertical coverage of the measurements.* |
|---|---|

Line 290: I don't follow why data from the early part of the month aren't included? Please explain why here.

Response: Regarding the intra-day comparison between the model and the measurement, seven consecutive days were analyzed in Sect. 4.2.3. Because this was the longest stretch of consecutive measurements, the impact of intra-day signals could be assessed. Then the rest of the days in August has been shown in the Sect. S7 of the supplement, and this explanation has been added in the content.

| Lines 330∼333 | *In order to obtain a more detailed view on how the model behaves at higher temporal resolution, the longest stretch of consecutive measurements in the campaign (i.e., from 16 to 22 August ) are analyzed here. Figures 5 & 6 show the daily curves of $XCO^S_{2,sla}$ and $XCH^S_{4,sla}$ at five sites for these 7 consecutive days against the corresponding modeled values with hourly temporal resolution.* |
|---|---|

Re nighttime columns: It is not clear to me how you define a nighttime slant column concentration. Since the sun is below the horizon, obviously your zenith angles are negative. Additionally, your averaging kernel matrix is based on retrievals from the day time so it is not clear how you extrapolate it, and I don't think it makes sense to do that if you did. I would clarify what exactly you did here and I note that line 298 ("pressure weighted as a proxy") is very unclear.

Response: As explained at the beginning of Sect. 4.3.2 in the manuscript ('Comparison between model and measurements: intra-day concentrations'), the model-measurement comparisons of the total column concentrations for GHGs are only made during the daytime, from approx. 6:00 UTC to 17:00 UTC. That is, the calculation of the modelled slant column concentrations are only implemented during the daytime. These modelled slant column concentrations are referred to as $XCO^S_{2,sla}$ and $XCH^S_{4,sla}$. To better understand the model performance over the whole day, and the impact of nighttime signals on the model-data mismatch, the pressure-weighted total column concentrations along the perpendicular column (i.e., $XCO_2$ & $XCH_4$) are calculated based on the Eq. 1 of the manuscript. We have rephrased four parts in the manuscript for clarity as follows:

| Lines 275∼277 | *Simulated concentration fields of $CO_2$ and $CH_4$ used for the model-measurement comparison in this study must therefore be aggregated along the slant columns from the ground to the sun* *during the period of the available measurement dates.* |
|---|---|
| Lines 298∼300 | *When comparing the simulations to the measurements, the simulated $CO_2$ and $CH_4$ concentration profiles along the slant column during the period of the available measurement dates are aggregated to the AK-smoothed column concentrations ($XCO_{2,sla}^S$ and $XCH_{4,sla}^S$) by using Eq. 3.* |
| Lines 337∼339 | *As mentioned in Sect. 4.2.1, the modelled and observed slant column concentrations ($XCO_{2,sla}^S$ and $XCH_{4,sla}^S$) used for the model-measurement comparisons are smoothed using the SZA-dependent AK based on Eq. 3.* |
| Lines 338∼339 | *Figures 5 & 6 also illustrate the contributions to the total column concentrations of $CO_2$ & $CH_4$ ($XCO_2$ & $XCH_4$) from different tracers in the model* *throughout day and night, which are calculated based on Eq. 1.* |

Line 422: The upwind site can't be used as a relative background if there are sources upwind of it that are still unmixed. In that case the upwind will see a stronger signal from those sources than the downwind one and they wouldn't be removed in a gradient. This seems to be what you are seeing with your data so you should mention this caveat here.

Response: Thank you for this insightful comment. We have added this caveat to both "preparation of DCM" and the discussion of concentration gradients.

| Lines 471∼472 | *In addition, in cases where there are emitters located upstream of an upwind site, the presence of the strong local signal prevents it from being used as a background site.* |
|---|---|
| Lines 529∼532 | *As a large methane sink over the city is not expected, the most likely cause for this phenomenon is that* *emission sources located upstream of an upwind site (i.e., somewhere to the northeast or east of the Garching and Markt Schwaben stations in the case with NE/E winds) are missing or underestimated in the initial emission inventory.* |

Line 423: STILT footprints have a magnitude and the footprints generally fall off as you move away from the instrument. So two footprints can overlap within their 90% contours but the instruments can have different measurements since one will be more strongly influenced by some sources (and see a higher peak) than the other. I would keep this in mind when discussing footprint overlap and gradients and think that looking at the difference in magnitude of the footprints could be more quantitative.

Response: Thank you for your valuable suggestion, we agree with your point. Based on this explorative study, we expect to be able to track down unknown or underestimated emitters more quantitatively, albeit this would require a longer observation record as well as some refinements to the modelling approach. Following your suggestion, we have extended the content in Sect. 5.5 ("Localizing unknown/underestimated emission sources").

| Lines 566∼569 | *However, with a longer observation record and refinements to the modelling approach, we see the potential to track down strong emitters of GHGs.* *It should be noted, however, that accurate estimation of unknown or underestimated sources needs to be performed using a combination of observations and a quantitative footprint analysis.* |
|---|---|

Line 489: You should include a time series of XCH4 gradients similar to figure 9. Without one, it is quite difficult to follow your logic in this section.

Response: We have included a time series of the XCH4 gradient similar to that in Fig. M9 in the supplement. The details can be found in Sect. S13 of the supplement.

Line 499: It is also good to note that your network does not isolate emissions from Munich which is what it was designed to. I think it would be useful to discuss how performing gradients on a 10's of km scale and in a complex source region can complicate isolating signals from an area of interest. This type of limitation is important for other groups.

Response: Thanks for this suggestion. We are also aware of both limitation of our model framework and the measurement network, especially for the isolation of $CH_4$ emissions over a complicated source area. The results of this research could serve as a helpful reference for other organizations seeking to identify the most effective tools and analytical methods for monitoring urban anthropogenic signals. Following this suggestion, we have mentioned these limitations in the conclusion of this study:

| | |
|---|---|
| Lines 583∼588 | *Despite the continuous total-column measurements surrounding the city center, this analysis highlighted the challenge of extracting the anthropogenic signal from such data. Even though our model was not able to fully reproduce the measured gradients in $CH_4$ over this complicated source region, this exploratory application enabled us to identify unexpected signals in the measurements and to roughly delineate the potential uncertain source regions in the inventory. The outcomes of this study may provide guidance for other groups considering the optimal instrumentation and analysis frameworks for measuring urban anthropogenic signals.* |

Line 519: Your paper discusses two distinct modelling frameworks under WRF-GHG and STILT but you do not compare the approaches in terms of their benefits and drawbacks and when one might be more appropriate than the other. I think incorporating this will add a lot to your paper and be useful for other groups.

Response: Thanks for this comment. Generally, WRF-Chem and STILT are two different types of models and are rather complementary than comparable. WRF-Chem is an Eulerian model, which combines weather prediction with tracer transport and can reproduce or predict high-resolution meteorological and concentration fields on a three-dimensional grid as a frame of reference. STILT, in contrast, is a stochastic Lagrangian (moving frame of reference) model simulating transport with pre-computed (assimilated) meteorological fields as input, whose calculations are offline and computationally efficient. Lagrangian models are efficient also because they need only resolve the meteorology along the course of a given advected air parcel. The two models are linked in this study because the meteorological output of WRF-Chem is used to drive the particle transport model STILT. Thus, STILT is simply used to derive the footprints, or areas of influence, of the measurements, using the same meteorological fields (Pillai et al., 2012). While the "direct" extraction of footprints from WRF outputs is theoretically possible with the adjoint form of WRF, it is considerably more computationally costly and complicated. While there are some differences regarding the representation in the vertical transport between WRF and STILT, we believe that any inconsistencies are irrelevant for the conclusions of this study. Therefore, using STILT driven by WRF meteorology for the purpose of this study is simply an efficient approach. Based on our model-measurement comparison of meteorological fields, particularly winds, we believe that the modelled wind fields from WRF are adequate for both Eulerian and Lagrangian.

In addition, we have added two tables (Table S5 & S6)in Sect. S11 of the supplement, one for the comparison between the two models with respect to their features, benefits and drawbacks (see Table R1), and another for the basic set-up of the models used in our study (see Table R2).

Re your comments about unknown emissions: The TNO inventory you use seems to include point sources, agriculture, and waste according to your van der Gon et al. (2019) reference. Additionally, it seems like the sources you plot in figure 10 fall on regions with emissions. So it might not be correct to say there are missing or unknown sources (e.g., line 365, 394, and in the abstract) that are responsible for your discrepancies without work to ensure that the sources in the inventory are not responsible for the mismatch you see.

Response: Thank you for the comments. As demonstrated in Sect. M2, the TNO_GHGco_v1.1 used in our

Table R1: Summary of two models used in this study

| Name | WRF-Chem (Peckham, S., 2017) | STILT (Fasoli, B., 2018) |
|---|---|---|
| Type of Model Framework | Eulerian framework | Lagrangian framework |
| Background | An extended version of WRF coupled with Chemistry, including transport of aerosol, NOx, GHG, etc. | An extension of HYSPLIT (The Hybrid Single-Particle Lagrangian Integrated Trajectory) model to simplify atmospheric transport modelling workflows and improve accuracy. |
| Usage of Model in this study | Simulation of spatial and temporal distributions of tagged trace gases, driven by the WRF modelled meteorology. | Simulation of transport of an ensemble of air parcels at a receptor location backward in time. |
| Input/Driver | Global reanalysis coarse meteorological databases, gridded emission fluxes and global background concentration profiles. | Global or regional meteorological fields. Vertical profiles of the horizontal and vertical wind components as key drivers. |
| Mechanism | Dynamical downscaling by solution of differential equations for the meteorological variables with transport/chemistry added using a tracer approach. | Transport of particles is directly followed by using a combination of mean winds from inputs with stochastic fluctuations based on a Markov process (turbulent motions). |
| Products | Finer meteorological fields and its derivative products, e.g., spatial and temporal distribution of $CO_2$. | Particle trajectories and gridded surface flux footprints (i.e., sensitivities to upstream surface emission fluxes). |
| Benefits | 1. Modelled outputs advanced in both spatial and temporal resolution. 2. Simulation of trace gases simultaneously with the meteorology, without time interpolations. 3. More realistic representation of the atmosphere and numerically more consistent, with the same grid structure of the meteorology. 4. The outputs assessed and used to interpret a variety of types of observations, also from satellites. | 1. More computational efficiency and easy extraction of footprints. 2. Concentration enhancements at the receptor which can be obtained by convolving the outputs (footprints) with emission inventories. |
| Weaknesses | 1. High computational requirements. 2. Less flexibility for conducting ensemble modeling and extracting the footprint information. | 1. Sensitivity of particle transports to the accuracy of meteorological input fields. 2. Turbulent transport in STILT is reproduced by following a stochastic process (Markov chain), which is highly sensitive to the vertical velocity variance and the Lagrangian time-scale (Pillai et al., 2012). |
| Relationship between models | WRF derived meteorological input fields are commonly used to drive STILT. | |

study contains point and area sources, classified into fourteen sectors. Figure. S2 depicts the vertical profiles for the point sources from different sectors in this inventory, and Tables S2&S3 show the aggregation of emission categories for $CO_2$ and $CH_4$ to our model.

Thanks for this correction. Not only could the 'missing' or 'unknown' sources lead to an underestimation of concentration gradients between down- and upwind sites, but also the underestimation in the magnitude of emission fluxes from the inventory could contribute to this model-measurement bias. We have rephrased all the parts in which the word 'underestimated' was missing in the original manuscript.

Table R2: Basic set-up of two models used in this study

| Name | WRF | STILT |
|---|---|---|
| Horizontal Resolution | 400 m × 400 m | 0.01 ° × 0.01 ° (resolution of footprints) |
| Vertical layers | 45 | 13 (particles' release height) |
| Meteorological inputs | ERA5 (37 km × 37 km), Emission fluxes from TNO-MACC, background concentration profiles from CAMS. | WRF output from the middle domain (D02; 1 km × 1 km). |
| Final products | Spatial and temporal distributions of meteorological fields and tagged trace gases (i.e., $CO_2$ and $CH_4$). | Gridded pressure-weighted column footprints for five MUCCnet sites. |

| Title of Sect. 5.5 | *Localizing unknown/underestimated emission sources* |
|---|---|
| Lines 441~442 | *...and most importantly for our attempt to locate unknown or underestimated $CH_4$ emissions (Sect. 5.5)* |
| Lines 467~468 | *Understanding these differences is a key prerequisite for determining the location of potential unknown or underestimated GHG sources based on noteworthy signals in the downwind-upwind concentration gradients.* |
| Lines 474~475 | *...the small non-overlapping parts are potential locations for unknown or or underestimated GHG emitters and sinks to be pinned down.* |

Additionally, if there are issues with the multinational inventory, it may be good to reference and discuss work by groups in Toronto (Pak et al., 2021) and California (Cararnza et al., 2018 & Mareklein et al., 2021) to create a fully resolved and detailed methane inventories in preparation for top-down monitoring. Otherwise, see my comment about considering other explanations for your discrepancies.

Thank you for the recommendation of the studies related to regional emission inventories. Bottom-up multinational emission inventories of $CH_4$ are generally compiled by scaling emissions using activity data and emission factors, which results in relatively large uncertainties (Bergamaschi et al., 2022). For the TNO_GHGco_v1.1 emission inventory used in this study, its point source information was collected on the location of power plants, large industrial installations, oil and gas production sites, airports and waste treatment locations (e.g. landfills), mostly from the E-PRTR (European Pollutant and Transfer Register) database. The data are valid for 2015. Therefore, in addition to the uncertainties due to the quantification of emissions in the inventory mentioned above, inconsistencies between emission information collected in 2015 or even earlier and actual emissions during the study period in 2018, could result in differences. Despite having chosen a high-resolution, state-of-the-art emission inventory, these uncertainties could contribute to the model-measurement differences.

In the recommended studies, the authors describe methods to generate/optimize customized inventories, which are quite impressive. However, the generation/optimization of emission inventories is not included in the range of our objectives for the current study. Nonetheless, our team is currently further developing inverse modelling approaches by using atmospheric measurement and a transport model (i.e., STILT) to optimize the emissions from the a-priori inventory. Two pilot cases have been published for Hamburg (Forstmaier et al., 2022) and for the city of Indianapolis (Jones et al., 2021). This is also being applied for Munich as part of the ICOS-Cities project.

Following your comments, we have added the causes of the uncertainties in the inventory to the manuscript as follows,

| Lines 549~556 | *In this study, the modelled contributions from human activities are initialized with the emission fluxes from the emission inventory TNO_GHGco_v1.1 for the year 2015. The multinational bottom-up emission inventory holds large uncertainties, due to the large variability in spatiotemporal distributions of $CH_4$ emissions from different sectors in different regions that have not yet been fully captured by the emission inventory (Bergamaschi et al., 2022), the disaggregation from annual emissions to hourly values using temporal profiles and the temporal inconsistency of emission information from 2015 or even earlier than the study period in 2018. This could result in missing or underestimated emissions in the inventory, as suggested by the measurements. After delineating the areas where the uncertain sources could be located, they were further pinpointed based on the updated database and local knowledge.* |
|---|---|

For their analysis of the methane gradients, the authors also conclude that there are missing spices on their inventory but they do not consider alternatives that can affect a column gradient. For example as elevation offsets (Hedilius et al., 2017), variability in the background signal at each instrument due to time lags in air reaching the boundary (Jones et al., 2021) or meteorological errors (c.f., Wu et al., 2018). I think the authors should consider these possibilities and others as well as the idea of missing data in their measurements.

Response: Thank you for this insightful comment, these points are quite helpful. The modelled meteorological fields could indeed bring about errors in the advection, which would contribute to the biases of absolute methane concentrations ($XCH_4$) and further to discrepancies in the methane gradients. Furthermore, due to lags in the time it takes for air to reach the boundary, the variation of the background signals at each instrument could be large. This is definitely a key point to be considered when the concentration gradients are used for inversions to optimize the inventories (as in Jones et al., 2021), but it presents significant complexities for our study and its implementation within the WRF-based framework. In addition to the emission-related causes that lead to the model-measurement biases of concentrations and their gradients, Hedilius et al.(2017) pointed out that non-emission factors (like the mixed layer height and topography) would further cause biased results. The importance of topography is verified as a significant factor in the variations of concentrations beyond the urban area. In our case, even though the elevations over our innermost domain (DO3, Munich) are rather consistent, i.e. around 550 m above sea level, the area around this domain's boundaries contains the complex topography of the Alps. It should be noted, that while it still plays a role, column measurements are less sensitive to mixing layer height than are in-situ measurements.

The causes discussed here could contribute to errors in the concentration and thus, the gradients. We have extended the discussion other causes which could contribute to the biases in the model-measurement gradients as such:

| Lines 540~544 | *In addition to the errors caused by the uncertainties in the initial emission inventory, other potential causes could contribute to errors in the concentration and thus, the gradients as well. The bias brought by the modelled meteorological fields can contribute to the bias of the modelled $XCH_4$, further to discrepancies in $\Delta XCH_{4,sla}^{S}$, by influencing on the advection (Wu et al., 2017). Further, our current DCM approach does not take the transport time into account. Moreover, factors such as the mixed layer height and topography could also introduce biases (Hedelius et al., 2017).* |
|---|---|

Re considering alternative explanations: with the preliminary data and model results, the authors suggest reasons for discrepancies they see in the model. However, in some instances, it seems like the authors do not justify why they settled on a particular explanation rather than another. For example, the authors attribute a bias in their $XCO_2$ model to the cams model used to provide boundary and initial conditions. However, they also mention that results at a sub-diurnal scale suggest a too-high net ecosystem exchange in their model and the authors suggest there is a flaw in their emission inventory with $CH_4$. How would the authors know if the cams model is

responsible for the mean bias they observe in the daily data?

Response: Thank you for these comments.

We would like to point out that to answer this question in detail, one needs to consider cases of $CO_2$ and $CH_4$ separately, as even though they are both simulated by the WRF and CAMS models, differences in their flux spatio-temporal patterns will lead to different biases when comparing to our measurements.

The modelled total column concentration of $CO_2$ ($XCO_2$) is made up of three parts, the background contribution (Model.(X)$CO_2$_BCK, see Fig. R4), the enhancements induced by human activities (Model.(X)$CO_2$_ANT) and biogenic activities (Model.(X)$CO_2$_BIO). Each of these components could contribute to the model-measurement discrepancy. As discussed in the manuscript, the bias in the model-measurement comparison of $XCO_2$ could be attributed to three main causes: i) overestimation of the modelled background concentration from CAMS, ii) errors in concentration enhancements brought by anthropogenic fluxes, and iii) errors in simulated biogenic fluxes.

[Figure]

Figure R4: Time series of the daily mean measured values over five sites of MUCCnet (black) and the averaged modeled $XCO_2$ from CAMS (red, CAMS.$XCO_2$_BCK) and WRF over D03 during the daytime (i.e., 6:00 UTC to 17:00 UTC). The modelled column concentrations are pressure-weighted means (see Eq. 1 in the manuscript). The error bars represent the standard deviation of the simulated values over D03 and over the five sites of MUCCnet. The orange curve represents the mean modelled column background concentration (Model.$XCO_2$_BCK). The green curve shows the averaged total column concentration (Model.$XCO_2$_BCK+Model.$XCO_2$_ANT+Model.$XCO_2$_BIO) and the blue curve shows the averaged column concentrations considering only the background and anthropogenic activities (Model.$XCO_2$_BCK+Model.$XCO_2$_ANT), without biospheric fluxes.

To understand the background-related cause in depth, we analyzed the variations and time series of CAMS itself, and compared the modelled and measured values. This part has been included in the edited supplement (see Sect. S14). As seen from the red and orange curves in Fig. R4, the day-to-day magnitude and variations in Model.$XCO_2$_BCK are mostly determined by its initialization (CAMS.$XCO_2$). For the simulations of background concentrations of tracer gas in WRF-Chem, it begins with initializing the 3-D field of the tagged tracer at the very beginning of the simulation cycle (i.e. 30th July in our study) and it is updated via the lateral boundary conditions using global tracer fields at a 3-hour interval (using CAMS fields for both). On the basis of Model.$XCO_2$_BCK (orange), the daily-mean total column concentrations (green) vary slightly with the positive anthropogenic fluxes (Model.$XCO_2$_ANT) and the carbon sink from biogenic activities during the daytime (Model.$XCO_2$_BIO). The mean bias between CAMS.$XCO_2$ and Obs.$XCO_2$ ($\pm$ its standard deviation) is 4.8 $\pm$ 0.7 ppm. Even though the overestimation of anthropogenic fluxes from the inventory and the uncertainty in the estimation of biogenic fluxes by the model could contribute to the model-measurement bias, this overestimation of CAMS overall plays a dominant role in the magnitude of the model-measurement bias of $XCO_2$. Gałkowski et al., (2021) & Tu et al., (2020) have supported this assumption that the CAMS (background) could cause a relatively large offset, rather than local emissions causing errors in excess of more than 3 ppm.

Moreover, a noteworthy dip can be observed in the model on 22 and 23 August (see the pink box in Fig. R4). This could be caused by the advection of air masses strongly influenced by photosynthetic uptake, coming into the domain from e.g., Italy, Slovenia, and Croatia, as has been discussed in Sect. 4.2.3 in detail.

[Figure]

Figure R5: Vertical profiles of (a)Altitudes, averaged modelled $CO_2$ over D03 on (b)16, (c)17, and (d)22, August 2018 from CAMS and WRF-Chem. The red curve represents the values from CAMS, and the others stand for our model results, with green for the total values, blue for the sum of the background and the human-related enhancements, and red for the background.

We also checked the vertical distribution of the model values from CAMS and WRF-Chem on 16, 17, and 22 August at 12 UTC (see Fig. R5). In general, the vertical distributions of CAMS $CO_2$ and Model $CO_2$_BCK are quite similar but slightly differ close to the ground level. This also indicates that the magnitudes and the vertical structure of background initialization of $CO_2$ (CAM.$CO_2$) play a decisive role in the modelled background (WRF.$CO_2$_BCK) and total concentrations. Furthermore, emissions caused by human activities (blue, Fig. R5) contribute to the total concentration (green) within the planetary boundary layer (PBL, below approx. 2 km). For the enhancements associated with biogenic activities (green curve), carbon sources from respiration contribute significantly to the total concentration of $CO_2$ near ground level, while air masses heavily influenced by photosynthetic uptake (with less $CO_2$) and coming from the outer domain play a key role at higher altitude, especially on 22 August. This could explain the dip on this date (see the pink box in Fig. R4). The animation of biogenic concentrations over D01 attached in the supplement provides a visual perspective of this phenomenon.

Owing to the relatively large bias of $CO_2$ brought in by CAMS, we considered using the model-measurement MB over all the measurement dates (i.e., 3.7 ppm) to "correct" the modelled values. This could help to see if the

model could reproduce similar variations to those seen by the measurements. These variations are determined by the modelled biogenic effects, initial emission fluxes from the inventory, the modelled advection of air masses influenced by human and biogenic fluxes, etc.

However, there is no significant model-measurement bias can be found in the daily-mean $XCH_4$ (cf. Fig. 3 (c) & (d) of the manuscript). Due to the quite weak biogenic activities of $CH_4$ in and around Munich (cf. Fig. 4 of the manuscript), the model-measurement bias of $CH_4$ is mostly caused by the uncertainties in human-related emissions.

Additionally, at line 302, can you quantify the background variability your model predicts? That information is useful for interpreting your gradients as any background variability between sites would remain in a gradient.

Thank your for the suggestion. We have added it to the manuscript as follows:

| Lines 347~349 | *In general, after subtracting the MB between* $XCO_{2,sla}^S$ *and the measured values over all the measurement dates (cf. Sect. 4.2.2), there is little difference in the column background concentrations among the five sites (black lines), with a mean and standard deviation of 404.8 $\pm$ 0.2 ppm over these 7 consecutive days.* |
| --- | --- |

Re statistics: the authors report the quality of their model results using the mean bias between the measurements and model, the root mean square error of the model and the Pearson's correlation coefficient. However, in a few places in the manuscript, the authors discuss the variability in their measurements and model (e.g., line 310, 342 and say to the effect that the model captures the variability in the measurements. However in these instances they refer to the mean bias or RMSE. This is somewhat misleading. Because the range of variation in their data is on the order of 1-4 ppm for $CO_2$ for example, a bias or RMSE of 1 ppm in the model is actually quite significant and wouldn't necessarily represent the model capturing the variation or performing well. In those cases the r2 which is scaled by the variance in the variables is more appropriate. I suggest that the authors rework their discussion of the fit of their model to reflect that.

Response: Thank you for your suggestions. Following your suggestions, we have added $R^2$ to the description and rephrased our content as follows,

| Lines 357~358 | *The modelled* $XCO_{2,sla}^S$ *(green +) reproduces the variability in the measurements (purple o) reasonably, with a RMSE of 1.33 ppm, a MB and its std of -0.79 $\pm$ 0.14 ppm, and a coefficient of determination ($R^2$) of 0.43, as it turns out.* |
| --- | --- |
| Lines 390~391 | *The modelled values show little diurnal variability at all sites compared to the measurements (RMSE: 6.7 ppb, MB $\pm$ std: -3.3 $\pm$ 5.9 ppb, and $R^2$: 0.31).* |

Re negative gradients in both directions: it is quite surprising that you find negative gradients in CH4 in both directions. Does that indicate that emissions from Munich itself are small on the days you measure or just that it isn't captured by your network? What are the implications of its emissions being missed during "good" measurement days in terms of your network design?

Response: The question is valid. We see the negative gradients in both directions for three test days. This means that the measured signals at the upwind sites are higher than the ones at the downwind site under northeasterly and northwesterly prevailing winds. This indicates the emissions outside of the city Munich over upstream of the upwind sites are missing in the initial emission inventory for both northwest and northeast sites. The causes of the unknown or underestimated sources in the inventory have been discussed above in this revision (see page 10 of this revision). Thus, in our network design, the implication here is that more emission sources are likely located over the upstream of the upwind sites in reality (the areas delineated in Fig. M10), while these are missing or underestimated in the initial inventory.

Line 318 and 342 and Section 4.2.3: Please clarify why your model XCO2 is consistently lower than your measurements. Since you removed an overall bias here I would expect the mean bias would be around 0. I also don't understand why the MB isn't 0 since you said above you subtracted it out. In Sect.4.2.3., it is not clear if the model comparison you're making in this section has a mean bias removed. Please clarify near the start of the section how exactly the data is treated.

Response: Thank you for this comment. For $CO_2$, to eliminate the bias which could be mostly caused by the overestimation of background and to better observe the day-by-day variations in $XCO_2$ (see Fig. M4), we subtract the MB over the entire available measurement period (i.e., the 15 dates shown in Fig. M4; 3.7 ppm) from the modelled values for all sites and for each available measurement date. In the manuscript, we chose to show 7 continuous days (from 16 to 22 August) as our key study period (see Sect. M4.3.2), while the rest is included in the supplement (see Sect. S7). Therefore, the MB discussed in of Sect. M4.2.3 is the remaining mean bias over these 7 consecutive days, after correcting with the value derived from the full 15 days. To avoid confusion in Sect. 4.2.3, we have added the following sentence:

| Lines 342~346 | *As described in Sect. 4.2.2., a MB of 3.7 ppm in $CO_2$ has been found over all the available measurement dates (see Fig. 4), which is defined to be the difference between the smoothed and measured daily mean $XCO_2$ and the modelled values. To eliminate the bias (too high modelled background $CO_2$) and focus on the model-measurement differences due to other causes, this MB is subtracted from the modelled $XCO_2$ in the day-by-day model-measurement comparison for all sites and for each simulation date.* |
| --- | --- |

Re Meteorological validation: Wu et al. (2018) found that the boundary layer height can have an impact on column measurements in addition to the 3d winds. Are you able to assess how your model performs in terms of boundary layer height?

Response: WRF provides the planetary boundary layer height (PBLH) as an output parameter but we lack observations to assess the performance of the modelled PBLH. Figure R6 shows the variation of the WRF-modelled PBLH. The modelled PBLH varies from a few hundred meters (morning and night) to over one thousand meters (midday), spanning a range from the first vertical layer (morning and night) to the 23$^{rd}$ layer (noon).

[Figure]

Figure R6: Diurnal variation of PBLH on 16 August. The deep blue line represents the median modelled values for each hour and the shaded blue area shows the corresponding inter-quartile ranges of the modelled PBLH.

**Minor comments**

Line 5: suggest you change "and interpret" to "attempt preliminary interpretation" to reflect that your conclusions on the model measurement mismatch are still preliminary.

Response: Thank you for this valuable comment, it has been adapted. We have rephrased the sentence accordingly:

| Lines 4∼6 | *We set up a modelling framework using the Weather Research and Forecasting (WRF) model applied at a high spatial resolution (up to 400 m) to simulate the atmospheric transport of GHGs and* attempt a preliminary interpretation of *the observations provided by the Munich Urban Carbon Column Network (MUCCnet).* |
|---|---|

Line 10: I would alter or clarify the phrase "1 to 30 August" to clarify that you are only able to use measurements on a limited subset of days rather than the full month.

Response: Thanks for pointing this out. The measurement campaign in this study was operated continuously during the entire period, i.e., 1 to 30 August, 2018. Meanwhile, our WRF model results are provided over the entire campaign period. With the restriction of measurement conditions and data quality, the measurements are filtered, leaving a limited amount which can be used in the further model-measurement comparison. If here the "limited subset" was used in the abstract, it might lead to confusions, that is, only these selected days in which the observations and the model were performed in the study. To avoid the misleading mentioned in this comment, we have re-phased this sentence as follows,

| Lines 10∼11 | *The measurements were provided by the instruments of MUCCnet and the campaign was carried out from 1 to 30 August, 2018.* |
|---|---|

Based on your discussion, it seems that the $CO_2$ signals are not well captured by the model so I suggest you change this line to reflect the different results between $CO_2$ (poor match) and $CH_4$ (good match) in your WRF-GHG analysis.

Response: Thanks for this comment. The sentence has been rephrased as follows:

| Lines 12∼15 | *In general, the model is able to reproduce the measured slant column concentrations of $CH_4$ and their variability, while for $CO_2$, a difference in the slant column $CO_2$ of around 3.7 ppm is found in the model. This can be attributed to the initial and lateral boundary conditions used for the background tracer.* |
|---|---|

Because the variability in your measured $XCO_2$ is on the order of 1 ppm, I would note that the 3.7 ppm bias you observe is quite large and could be taken to indicate a poor model fit for your initial WRF-GHG fields.

Response: Indeed, the overall bias of 3.7 ppm of $XCO_2$ is large when compared to the amplitude of the variability measured. However, this bias was found to be rather constant over time (e.g., in Fig. R4), and as such does not impact the modelled gradients. This large offset can be explained by an offset in the initial and lateral boundary background $CO_2$ concentrations provided by CAMS. A detailed discussion related to the background tracer can be found on page 12-13 of this revision.

In your write up you say that some of the error in $CO_2$ may be attributable to flaws in your biogenic $CO_2$ flux but here you say it's due to the initial and background conditions. Please clarify this?

Response: Thank you for the comment. Following the detailed discussion in the $CO_2$ bias on page 11-13 of this revision, both the background and the biogenic fluxes could contribute to the bias in the slant column of $CO_2$. While the background fields are responsible for offsetting a rather large (and relatively constant) bias, the biogenic fluxes are found to account for mismatches in both the diurnal cycle and interdiel variability over the seven consecutive days analyzed in Sect. 4.2.3. We have added the bias in the modelled slant column concentration of $CO_2$ caused by biogenic activities in the abstract.

| Lines 14~15 | *Additional mismatches in the diurnal cycle could be explained by an underestimation of nocturnal respiration in the modelled $CO_2$ biogenic fluxes.* |
|---|---|

Your abstract is missing a discussion of your $CH_4$ results with WRF-GHG or STILT.

Response: Some mention was made of the $CH_4$ results with WRF-GHG and STILT in the original abstract (e.g. Lines 18-19: *Combining these footprints with knowledge of local emission sources, we find evidence of $CH_4$ sources near Munich that are missing or underestimated in the emission inventory used*), and the references to GHG concentrations did not always make it clear that both $CH_4$ and $CO_2$ were included in the study. This has now been made more explicit in previous edits to the abstract.

| Lines 12~15 | *In general, the model is able to reproduce the measured slant column concentrations of $CH_4$ and their variability, while for $CO_2$, a difference in the slant column $CO_2$ of around 3.7 ppm is found in the model. This can be attributed to the initial and lateral boundary conditions used for the background tracer. Additional mismatches in the diurnal cycle could be explained by an underestimation of nocturnal respiration in the modelled $CO_2$ biogenic fluxes.* |
|---|---|

Line 13: In your write up you say that you can't interpret the $XCO_2$ gradients because of the biogenic fluxes so you may want to add that in your abstract as the way it is written now makes it sounds like you do interpret them?

Response: Thank you for this suggestions and we have added the content in the abstract to interpret the $CO_2$ bias related to the underestimation of biogenic fluxes (see lines 14-15 of the manuscript, also shown above).

Line 25: adaption should be mitigation.

Response: Thank you for pointing it out, it has been corrected in line 28 of the manuscript.

Line 59: I think that you should also note that top-down emissions have uncertainty due to their own spatial and temporal representativeness (cf Vaughn et al., 2018) in addition to the other reasons you list. Those issues need to be carefully treated to interpret emissions from top-down.

Response: Thanks for this comment and we have extended the text accordingly:

| Lines 59~64 | *Inversion models still show considerable potential for improvement, owing to limited knowledge about the characteristics and spatial distribution of emission sources (e.g., missing or underestimated sources, inner-city traffic), uncertainties in background concentrations, and the difficulty of modelling transport in complex urban environments.* *Furthermore, emissions that are highly heterogeneous in time and space are challenging to be assessed using atmospheric measurements and models (e.g., Vaughn et al. (2018)).* |
|---|---|

Line 82: Jones et al. (2021) doesn't do a bayesian inversion using biogenic signals so I don't understand your reference here. They do an inversion for CH4 in an urban area.

Response: Thank you for this comment. Jones et al. (2021) is cited to support the phrase "Bayesian inversion model". This is why the citation is located directly after the phrase, instead of the end of the sentence. We are developing a $CO_2$ Bayesian inversion framework based on the structure described in Jones et al.(2021). Specifically, based on Jones' inversion framework and the biogenic products from WRF-GHG, we attempt to further develop an inverse modeling approach to estimate $CO_2$ surface emissions. This is an ongoing project and there are no publications available for citation at this time. We note that the expression in this sentence, especially the present perfect tense, is misleading, and we have changed it to the present progressive tense:

| Lines 86~89 | *For instance, highly-resolved meteorological fields can drive particle transport models (Fasoli et al., 2018). These Lagrangian footprints can then be used for inversion studies, similar to Heerah et al. (2021), who optimized dairy $CH_4$ emissions across the San Joaquin Valley using WRF-STILT inversions. Currently, an adapted Bayesian inversion model based on Jones et al. (2021) is being developed to infer anthropogenic $CO_2$ emissions, with the consideration of biogenic fluxes.* |
|---|---|

Line 115: not sure what you mean by morphological, you just used specific land use correct?

Response: Thank you for pointing it out and it has been rephased in the content.

| Lines 120~122 | *To better capture the urban landscape features and improve the urban model performance (Ching et al.,2018; Mughal, 2020), extra urban land use land cover categories are provided for the innermost domain (D03, area of Munich), which enables us to use the urban canopy multi-layer scheme in WRF (Brousse et al., 2016).* |
|---|---|

Line 119: refer to a specific part of the supplement.

Response: Here the reference is to Sect. S3 of the supplement, and we have added it in line 125 of the manuscript:

| Lines 125 | *More information regarding this procedure can be found in Sect. S3 of the supplement.* |
|---|---|

Line 120: Later in your write up you refer to the background signals as CAM so I would strongly suggest you introduce and use that acronym here for clarity. As written now, I was confused if IFS and CAM later on were different.

Response: Thanks for this comment. In this study, the IFS Cycle 45rl is used for the initialization of background concentration fields. This database is generated as a product of the global Copernicus Atmosphere Monitoring System (CAMS). CAMS makes use of the Integrated Foresting System (IFS) from the European Center for Medium-Range Weather Forecasts (ECMWF). The experiment ID of this product is 'gqpe'.

| Lines 128~130 | *The IFS Cycle 45rl is operated by the European Center for Medium-Range Weather Forecasts (ECMWF) as part of the Copernicus Atmosphere Monitoring Service (CAMS). The IFS cycle 45rl is referred to as CAMS for simplicity.* |
|---|---|

Line 188: How do you treat wind direction differences that occur across the cut in wind directions (i.e., 179 and -179 are only 2 degrees apart but would be 358 degrees with a simple difference). Additionally, since you mention the standard deviation in the wind direction and since this is a proxy for stability which is important for transport modelling, it might be appropriate to include that as a panel.

Response: Thanks for your question. We do consider this situation when calculating the evaluation parameters

of wind direction. We treat them following the method presented in Jiménez et al., 2013. The difference $\Delta wd$ between the modelled and measured wind directions are defined as,

$$\Delta wd = \begin{cases} wd_{model} - wd_{obs} & \text{if} \quad wd_{model} - wd_{obs} \leq |180| \\ wd_{model} - wd_{obs} - 360 & \text{if} \quad wd_{model} - wd_{obs} > 180 \\ wd_{model} - wd_{obs} + 360 & \text{if} \quad wd_{model} - wd_{obs} < -180 \end{cases}$$

This definition assigns a positive (negative) difference to the wind direction when the modelled wind direction is rotated clockwise (counter-clockwise) with respect to the observations. The value of $\Delta wd$ ranges between -180 ° and 180 °. Thus, RMSE, MAE and other representatives errors can be calculated accordingly.

We have also added the content in the manuscript as follows,

| Lines 194∼195 | *The evaluation of wind directions are treated following the method presented in Jiménez and Dudhia (2013).* |
|---|---|

Figure 2: I would suggest you adopt an unambiguous date time format in your x axis as the mm/dd/yy is the standard in America but not elsewhere.

Response: Thanks for pointing this out. The date format has been changed to dd/MM/YY in Fig. M2.

Line 204: The reference to Hedilus et al. (2016) is incorrect. I would suggest Gisi et al. (2012) as the paper to refer to for the EM27/sun operating principles.

Response: Thanks for pointing it out, it has been amended.

| Lines 230∼231 | *By using the sun as a light source, the EM27/SUN measures near-infrared solar spectra (Gisi et al., 2012).* |
|---|---|

Line 205: the final product of the standard retrieval for the EM27/SUN is typically a total column average dry air mole fraction not an abundance so it might be better to change the word "abundance".

Response: Thank you for pointing this out. To avoid any ambiguity, we changed "abundance of $CO_2$ and $CH_4$ in column" to "column averaged dry-air mole fractions of $CO_2$ and $CH_4$".

| Lines 231∼233 | *In MUCCnet, the recorded interferograms are automatically transformed to spectra, converted to column averaged dry-air mole fractions (DMF) of $CO_2$ and $CH_4$ between the instrument and the end of the atmosphere in the direction towards the sun, and further uploaded to the official website of MUCCnet.* |
|---|---|

Line 205: Do you use GGG or PROFIT to fit your spectra? I would include that information and cite the relevant software.

Response: For the measurement campaign of 2018 discussed in this study, the GFIT GGG-2014 algorithm was applied for the retrievals, while currently, the PROFIT and GGG-2020 algorithms are used in MUCCnet. We have clarified this part in the manuscript.

| Lines 233∼235 | *The retrieval algorithm GFIT GGG-2014 (Wunch et al., 2015) was applied during the measurement campaign of 2018, while currently MUCCnet is using the PROFIT (Hase et al., 2004; Frey et al., 2019) and GGG-2020 algorithms (Laughner et al., 2023).* |
|---|---|

Line 218: I don't think that Vogel et al. is the appropriate citation for saying the FTS is influenced by meteorology. Their work is more of an application rather than an error or bias assessment and Gisi et al. (2012) or Hediulus et al. (2016) may be more appropriate. If you meant that you're adopting their measurement screening it might be better to say "We screened the measurement days following Vogel et al. (2018)."

Response: Thank you for pointing this out, and for the valuable suggestion. Obviously, the EM27/SUN could only operate in sunny daylight conditions, since it uses sun light as sources. We agreed with your suggestion and decided to delete this citation here (see lines 246-247 of the updated manuscript).

Line 219: I think changing the phrase "measurement performance" to something like measurement quantity may be a good idea because the number of spectra is the only aspect of performance that you seem to assess.

Response: Thanks for your suggestion and we decide to change the phase 'measurement performance' to 'characteristics of measurement days'. This is because in Table. S4, we include not only the quality of the measurements (i.e., number of observations, overall quality ranking, temporal coverage), but also information on the wind. Moreover, this phase is also used in Hase et al. (2015). The head of Table. S4 can be found in Fig. R7, and the change to the text is shown with lines 248-252 in a previous response. In the supplement we changed the heading of the section to "Measurement days – data-quality characteristics", and the introductory sentence was changed to match the new table caption.

| Lines 249∼252 | *By assessing the main characteristics of the measurement days during the campaign, we selected fifteen days in total with good measurement conditions (i.e. with a quality level better than '++', cf. Table S4 and Sect. S6 of the supplement) to make the model-measurement comparison: 4-6, 9, 11, 16-22 and 27-29 August, 2018.* |
|---|---|

**Table S4.** Summary of day-by-day main characteristic of measurement days from 1 to 30, August in 2018 at our five measurement sites, including the number of measurement points for each site, overall data coverage for each measurement date (with the classifications from poor to excellent: +, ++, +++, ++++) based on the available observations, averaged wind speeds during the day time, and wind directions at the ground level obtained from the LMU stations (Hase et al., 2015; Vogel et al., 2019)

| Date | Quality | Number of Observations | | | | | Wind Speed | Wind Direction |
|---|---|---|---|---|---|---|---|---|
| | | Garching (North) | TUM (Center) | Höhenkirchen (South) | Markt Schwaben (East) | Weßling (West) | | |
| 20180801 (Wed) | + | 15 | 127 | 0 | 0 | 0 | 2.56 | W-N-E |

Figure R7: The new head of Table. S4 in the supplement.

Line 246: I don't understand why you cite Borsdorff et al here. If I understand your write up correctly you use the methodology in Zhao et al. and they don't mention using methods explicitly from this paper.

Response: Thanks for this question. In line 246 where Borsdorff et al. (2014) is cited, we would like to point out that AK stands for the altitude-dependent column sensitivity. Even though the purpose of Borsdorff et al. (2014) is to show a new algorithm as an extension to calculate total column AK, the relation between AK and altitude, even under different cloud fractions is discussed.

Line 263: I'm not sure if the reference to Tu et al. is right, it seems like they used a different cut off and if you just meant to refer to the fact that the retrieval becomes worse at higher air mass, Gisi et al. (2012) may be more appropriate.

Response: Thanks for this comment. You are right that Tu et al. (2020) uses another cutoff which is SZA > 80 °. This paper is cited because they stated that a cutoff was applied to filter out measurements, in order to reduce uncertainties associated with spectra recorded at high air masses (Sect. 2 of Tu et al., (2020)). Gisi et al. (2012) used a cutoff of 70 °. Following your suggestion, we have added 'Gisi et al. (2012)' in Line 263 as you suggested (see lines 295-297 of the updated manuscript).

| Lines 297∼298 | *Specifically, to reduce uncertainties caused by high air masses, measurements are discarded when they are observed at SZA larger than 75 degrees (Tu et al., 2020; Gisi et al., 2012).* |
|---|---|

Figure 3: For clarity I think it would be better to use different colour maps for the different locations and to represent dates in your scatter plots. Someone glancing at the plots could be confused as to what your scatter plot colours refer to (time rather than site).

Response: Thank you for your suggestion. For the scatter plots in Fig. M4, the color map represents the date, following its color map on the right-hand side. To avoid confusion when people glance at the plot, we have changed the colors used to represent different sites in Fig. M4(a)&(c), which do not belong to the rainbow color map. In addition, we moved the legend for (a)&(c) to the place below panel c, making the legend visible.

Line 269: It is unclear what you mean by "We have considered the limited measurement period"

Response: Thanks for pointing this out. Here we would like to show that we take the limited measurement period of the day into account, i.e., from around 6 am to 5 pm, when calculating the modelled daily-averaged concentrations. Specifically, we average the modelled concentration values over the measurement period of the day, instead of a simple daily average. To avoid confusion, we improved this sentence as follows:

| Lines 305∼306 | *When producing daily-averaged modelled values, we have considered the limited measurement period on each day, that is, from around 6:00 UTC to 17:00 UTC.* |
|---|---|

Line 292: Do you mean the unselected days?

Response: Thanks for this comment, but in this sentence, we still mean the rest of the selected days. As stated in Sect. 4.1 of the manuscript (i.e., lines 249-251), in total we selected fifteen days with good measurement conditions (with a quality level better than "++", c.f. Table S4 and Sect. S6 in the supporting information). Then seven consecutive days were featured in the manuscript (i.e., "selected"), and the rest of the days with good data (i.e., "unselected") are found in the supplement (see Sect. S7 of the supplement).

Line 308: I would note here that the period you refer to is the time you are actually measuring.

Response: Thank you for the note, we have edited it in Line 308 as follows:

| Lines 355∼356 | *There is no obvious difference between the modelled values with and without smoothing during the daytime from around 6:00 UTC to 17:00 UTC, which covers most of the period of the day during which measurements can be made.* |
|---|---|

Line 310: If you're talking about variability, an r2 might be more appropriate than the mean bias. Since you already removed some of the bias with your diurnal average analysis, it is not very surprising that the bias is low.

Response: Thanks for your suggestion, we have added the $R^2$ in the manuscript for both $CO_2$ and $CH_4$ (see lines 358 & 391). This has been discussed in the general comments above.

Line 310: Additionally, given that the variability in your measurements is on the order of 1 ppm, a bias of .8 ppm could be taken as being quite large.

Response: We do agree with your point and one of the key reasons for this big MB is caused by the discrepancy between the model and measurement for 22 August. This cause is mentioned in the explanation of the $CO_2$ bias in the general comments (see Pages 11-13 of this revision) and has been discussed in depth in lines 368-375 of

the updated manuscript.

Line 316: It would be useful and add to your argument to quantify the size in terms of ppm that this respiration effect might have on your data.

Response: In the WRF-GHG output, we could only obtain concentration fields caused by both photosynthesis and respiration effects as a whole with a unit of ppm. That is, when estimating the biogenic related concentration enhancements, the biogenic fluxes do not partition into GEE and RES. As photosynthesis is not active at nighttime, the biogenic fluxes is entirely contributed by night respiration. We could obtain the concentration enhancements induced by respiration effect (ppm) at nighttime. Gourdji et al. (2021) showed night-time daily mean NEEs over the year provided by five different versions of VPRMs. As photosynthesis is not active at nighttime, the biogenic fluxes is entirely contributed by night respiration. The modelled NEE from the traditional VPRM which is the same as what used in this study, varied from 1 (in winter) to 3 (in summer) $\mu\mathrm{mol}/(\mathrm{m}^2 \cdot \mathrm{s})$, while the values from the improved VPRM can be over 6 $\mu\mathrm{mol}/(\mathrm{m}^2 \cdot \mathrm{s})$ in summer. This could provides us with an estimate for such underestimations due to the underestimated respiration fluxes in the traditional VPRM. We have added this information in the discussion of the manuscript.

| Lines 363~364 | *Gourdji et al. (2021) found the differences of RES at nighttime in summertime between the improved and traditional (use in this study) VPRMs can reach more than 3 $\mu\mathrm{mol}/(\mathrm{m}^2 \cdot \mathrm{s})$, depending on vegetation types.* |
|---|---|

Line 375: if you mean accounting for differences in the wind vectors with height I think you should say accounting "for wind shear" rather than "differences in wind shear".

Response: We really thank you for pointing this out and we have edited it in the text (see line 424 of the updated manuscript).

Line 408: In your write up about WRF-GHG, you note that you were careful to use realistic release heights for your sources. In STILT, sources are assumed to be in the lower half of the boundary layer so saying surface emissions could be a little confusing.

Response: Thanks for this remark. Surface emission fluxes are well known to be close to the ground and generally released from the emitters near the surface. This is shown in Fig. S2 of the supplement). Specifically, the model provides us with the sensitivity of the analyzed slant columns to emissions in lower half of the PBL. To avoid confusion here, we have re-phased the sentence as follows,

| Lines 457~458 | *Specifically, the model provides us with the sensitivity of the analysed slant columns to emissions in lower part of the planetary boundary layer height.* |
|---|---|

Line 417: Jones et al. used NCEP pressure weights. Do you do the same or use your WRF field pressure weights?

Response: In our study, we followed the calculation stated in Jones et al. (2021) to obtain the pressure weights used for the calculation of column footprints (see Sect. S1 of Jones et al. (2021)).

Figure 8: red curve missing in panel c.

Response: The reason for the absence of the red curve in panel C is the lack of measurement data in the IGRA. An explanation for the missing panel has been added in the title of Fig. M3.

Line 460: See note about consolidating your wind comparisons.

Response: Thank you for this valuable suggestion. As discussed in the general commend (page 3), we have

moved the model-measurement comparison of the two radiosondes to Sect. 3 of the manuscript, which can be found on pages 8-9 of the manuscript.

Line 485: The underestimate is only in some sites in figure 9 so specify that.

Response: Thanks for this suggestion and we have specified the sites with the underestimation of modelled gradients in the manuscript.

| Lines 523∼525 | *However, the modelled and measured concentration gradients show some differences, e.g., the modelled concentration gradients between Höhenkirchen and Weßling before 10:00 UTC on 22 August were underestimated compared to the observations.* |
| --- | --- |

Line 601: Is there a link to this reference.

Response: The link to reference has been added.

Line 641: Link to reference?

Response: The link to reference has been added.

Figure 10: include natural gas pipeline in legend.

Response: It has been included in the updated version of Fig. M11 in the manuscript.

Line 715: link to reference and DOI for dataset?

Response: This is a report from the EU project CHE, without a DOI. The link to this report has been added in the reference.

SI Table S4: you mention the wind speed variability in your text but don't include it here.

Response: Thanks for this comment. Indeed, Table S4 at the moment only shows averaged wind information based on WS30 and WD30. We intended to have this comment on variability as a qualitative assessment which the reader will understand looking at the figure included in the main paper, and would indeed tend to leave it at this level. If the referee is of another opinion, we will however be happy to re-iterate it.

SI Figure S11: It appears to me that the 18th and 19th (for some instruments) also have a visual overlap in the footprints but are excluded. Can you explain your reasoning for the exclusion more? Additionally, why are the early month days excluded from the footprint analysis

Response: Thanks for this question. The cause why 18 and 19 are excluded in the discussion is that not all sites for these two days meet the criteria to be selected as either downwind or upwind sites based on Table M1 under the wind conditions of that day. In classical DCM, the air masses should theoretically pass by the upwind site and further arrive at the downwind site with a relatively stable wind, after travelling through an urban area in which most of emissions are located. As shown in Fig. S14(c), under such a prevailing wind of (i.e., northeasterly), Weßling (West) should be selected as the downwind sit for the calculation of concentration gradients. But some part of the upstream at this downwind site did not overlap the upstream of the other three upwind sites, shown in Table M1. Specifically, some air does not pass the upwind sites (mainly from the south/southeast region of Munich) and arrives at Weßling together with other airs that pass the upwind sites. This is also the case for 17 and 20 August (see Fig. S14(a)&(d)).

In terms of 18 August, under the prevailing winds (northwesterly, see Table S4 & Fig. S10), the upwind site is supposed to be Weßling (west), while the other three sites should be the downwind sites based on Table M1. But as shown in Fig. S14(b), the in-flowing airs reaching Weßling (red line) are mainly from the northern region of

Munich, and could not further pass the other three downwind site. This detailed explanation for the exclusion has been added to the supplement (see Sect. S10).

**References**

Bergamaschi, P., Segers, A., Brunner, D., Haussaire, J.-M., Henne, S., Ramonet, M., Arnold, T., Biermann, T., Chen, H., Conil, S., Delmotte, M., Forster, G., Frumau, A., Kubistin, D., Lan, X., Leuenberger, M., Lindauer, M., Lopez, M., Manca, G., Müller-Williams, J., O'Doherty, S., Scheeren, B., Steinbacher, M., Trisolino, P., Vítková, G., and Yver Kwok, C.: High-resolution inverse modelling of European CH$_4$ emissions using the novel FLEXPART-COSMO TM5 4DVAR inverse modelling system, Atmos. Chem. Phys., 22, 13243–13268, 10.5194/acp-22-13243-2022, 2022.

Carranza, V., Rafiq, T., Frausto-Vicencio, I., Hopkins, F.M., Verhulst, K.R., Rao, P., Duren, R.M. and Miller, C.E.: Vista-LA: Mapping methane-emitting infrastructure in the Los Angeles megacity. Earth System Science Data, 10(1), pp.653-676., 10.5194/essd-10-653-2018, 2018.

Dietrich, F., Chen, J., Voggenreiter, B., Aigner, P., Nachtigall, N. and Reger, B.: MUCCnet: Munich urban carbon column network. Atmospheric Measurement Techniques, 14(2), pp.1111-1126, 2021.

Fasoli, B., Lin, J. C., Bowling, D. R., Mitchell, L., and Mendoza, D.: Simulating atmospheric tracer concentrations for spatially distributed receptors: updates to the Stochastic Time-Inverted Lagrangian Transport model's R interface (STILT-R version 2), Geosci. Model Dev., 11, 2813–2824, 10.5194/gmd-11-2813-2018, 2018.

Gourdji, S., Karion, A., Lopez-Coto, I., Ghosh, S., Mueller, K. L., Zhou, Y., Williams, C. A., Baker, I. T., Haynes, K., and Whetstone, J.: A modified Vegetation Photosynthesis and Respiration Model (VPRM) for the eastern USA and Canada, evalu- ated with comparison to atmospheric observations and other biospheric models, Earth and Space Science Open Archive, p. 50, 675, 10.1002/essoar.10506768.1, 2021

Forstmaier, A., Chen, J., Dietrich, F., Bettinelli, J., Maazallahi, H., Schneider, C., Winkler, D., Zhao, X., Jones, T., van der Veen, C., Wildmann, N., Makowski, M., Uzun, A., Klappenbach, F., Denier van der Gon, H., Schwietzke, S., and Röckmann, T.: Quantification of methane emissions in Hamburg using a network of FTIR spectrometers and an inverse modeling approach, Atmos. Chem. Phys. Discuss. [preprint], 10.5194/acp-2022-710, in review, 2022.

Hedelius, J.K., Feng, S., Roehl, C.M., Wunch, D., Hillyard, P.W., Podolske, J.R., Iraci, L.T., Patarasuk, R., Rao, P., O'Keeffe, D. and Gurney, K.R.: Emissions and topographic effects on column CO$_2$ (XCO$_2$) variations, with a focus on the Southern California Megacity. Journal of Geophysical Research: Atmospheres, 122(13), pp.7200-7215, 10.1002/2017jd026455, 2017.

Jiménez, P. A., & Dudhia, J.: On the ability of the WRF model to reproduce the surface wind direction over complex terrain. Journal of Applied Meteorology and Climatology, 52(7), 1610-1617, 10.1175/JAMC-D-12-0266.1, 2013.

Jones, T. S., Franklin, J. E., Chen, J., Dietrich, F., Hajny, K. D., Paetzold, J. C., Wenzel, A., Gately, C., Gottlieb, E., Parker, H., Dubey, M., Hase, F., Shepson, P. B., Mielke, L. H., and Wofsy, S. C.: Assessing urban methane emissions using column-observing portable Fourier transform infrared (FTIR) spectrometers and a novel Bayesian inversion framework, Atmos. Chem. Phys., 21, 13131–13147, 10.5194/acp-21-13131-2021, 2021.

Marklein, A.R., Meyer, D., Fischer, M.L., Jeong, S., Rafiq, T., Carr, M. and Hopkins, F.M.: Facility-scale inventory of dairy methane emissions in California: implications for mitigation. Earth System Science Data, 13(3), pp.1151-1166, 10.5194/essd-13-1151-2021, 2021.

Pak, N.M., Heerah, S., Zhang, J., Chan, E., Worthy, D., Vogel, F. and Wunch, D.: The facility level and area methane emissions inventory for the Greater Toronto Area (FLAME-GTA). Atmospheric Environment, 252, p.118319, 10.1016/j.atmosenv.2021.118319, 2021.

Pillai, D., Gerbig, C., Kretschmer, R., Beck, V., Karstens, U., Neininger, B., and Heimann, M.: Comparing Lagrangian and Eulerian models for $CO_2$ transport – a step towards Bayesian inverse modeling using WRF/STILT-VPRM, Atmos. Chem. Phys., 12, 8979–8991, 10.5194/acp-12-8979-2012, 2012.

Peckham, S., Grell, G. A., McKeen, S. A., Ahmadov, R., Barth, M., Pfister, G., Wiedinmyer, C., Fast, J. D., Gustafson, W. I., Ghan, S. J., Zaveri, R., Easter, R. C., Barnard, J., Chapman, E., Hewson, M., Schmitz, R., Salzmann, M., and Freitas, S. R.: WRF-Chem Version760 3.9.1.1 User's Guide, Tech. rep., National Center for Atmospheric Research, https://ruc.noaa.gov/wrf/wrf-chem/Users_guide.pdf, 2017.

Vaughn, T.L., Bell, C.S., Pickering, C.K., Schwietzke, S., Heath, G.A., Pétron, G., Zimmerle, D.J., Schnell, R.C. and Nummedal, D.: Temporal variability largely explains top-down/bottom-up difference in methane emission estimates from a natural gas production region. Proceedings of the National Academy of Sciences, 115(46), pp.11712-11717, 10.1073/pnas.1805687115, 2018.

Wu, D., Lin, J. C., Fasoli, B., Oda, T., Ye, X., Lauvaux, T., Yang, E. G., and Kort, E. A.: A Lagrangian approach towards extracting signals of urban $CO_2$ emissions from satellite observations of atmospheric column $CO_2$ (XCO_2): X-Stochastic Time-Inverted Lagrangian Transport model ("X-STILT v1"), Geosci. Model Dev., 11, 4843–4871, 10.5194/gmd-11-4843-2018, 2018.

---

## Author Comment (AC2)

The paper compares wind and EM27/SUN data at 5 sites taken as part of the Munich Urban Carbon Column Network in August 2018 with a ¿400m resolution WRF model with emissions. The goal is top down verification of CO2 and CH4 emissions, that are challenging given that these are long lived species that are influenced by long range transport and also local sources. The analysis is detailed analysis is presented well and valuable particularly in identifying conditions of uniformity ion regional air masses when a "gradient" method is explored, that may be useful operationally for top down verification. The paper should advance GHG verification strategies.

I do have the following questions and concerns that demand further clarifications by the authors:

We thank the anonymous Referee #2 for their time and valuable comments to improve this manuscript. The questions posted are addressed in point-by-point replies below. The referee's comments have been repeated in black. The authors' replies are marked in blue and the edited contents of the manuscript are documented in red in tables below each comment. Moreover, we set the numbers of the figures in this revision as 'R' plus the numbers (e.g., Figure R1), while the figures in the manuscript are numbered with 'M' plus its numbers (i.e., Figure M1).

1. A more careful explanation of the CO2 bias would be useful as it appears to be constant and obviously a statement that it cancels out.

Response: Thank you for this comment. We have extended our discussion of the $CO_2$ bias, adding to the supplement (see Sect. S14). In this study, we consider cases of $CO_2$ and $CH_4$ separately, as even though they are both simulated by the WRF and CAMS models, differences in their flux spatio-temporal patterns will lead to different biases when comparing to our measurements.

The modelled total column concentration of $CO_2$ ($XCO_2$) is made up of three parts, the background contribution (Model.(X)$CO_2$_BCK, see Fig. R1), the enhancements induced by human activities (Model.(X)$CO_2$_ANT) and biogenic activities (Model.(X)$CO_2$_BIO). Each of these components could contribute to the model-measurement discrepancy. As discussed in the manuscript, the bias in the model-measurement comparison of $XCO_2$ could be attributed to three main causes: i) overestimation of the modelled background concentration from CAMS, ii) errors in concentration enhancements brought by anthropogenic fluxes, and iii) errors in simulated biogenic fluxes.

To understand the background-related cause in depth, we analyzed the variations in the time series of CAMS itself, and compared the modelled and measured values. This has been included in the edited supplement (see Sect. S14). As seen from the red and orange curves in Fig. R1, the day-to-day magnitude and variations in Model.$XCO_2$_BCK are mostly determined by its initialization (CAMS.$XCO_2$). For the simulations of background concentrations of tracer gas in WRF-Chem, it begins with initializing the 3-D concentration field at the very beginning of the simulation cycle (i.e. 30th July in our study) and it is updated via the lateral boundary conditions from the global fields at a 3-hour interval (using CAMS fields for both). On the basis of Model.$XCO_2$_BCK (orange), the daily-mean total column concentrations (green) vary slightly with the positive anthropogenic fluxes (Model.$XCO_2$_ANT) and the carbon sink from biogenic activities during the daytime (Model.$XCO_2$_BIO). The mean bias between CAMS.$XCO_2$ and Obs.$XCO_2$ ($\pm$ its standard deviation) is 4.8 $\pm$ 0.7 ppm. Even though the overestimation of anthropogenic emission fluxes from the inventory and the uncertainty in the estimation of biogenic fluxes by the model could contribute to the model-measurement bias, this shows that the overestimation of CAMS overall plays a dominant role in the magnitude of the model-measurement bias of $XCO_2$.

We also checked the vertical distribution of the model values from CAMS and WRF-Chem on 16, 17, and 22 August at 12 UTC (see Fig. R2). In general, the vertical distributions of CAMS $CO_2$ and the modelled $CO_2$_BCK from WRF are quite similar but differ slightly close to the ground level. This also indicates that the magnitudes and the vertical structure of background initialization of $CO_2$ (CAM.$CO_2$) play a decisive role in the modelled background (WRF.$CO_2$_BCK) and total concentrations. Furthermore, emissions caused by human activities (blue, Fig. R2) contribute to the total concentration (green) in the planetary boundary layer (PBL, below approx.

2 km). For the enhancements associated with biogenic activities (green curve), carbon sources from respiration contribute significantly to the total concentration of $CO_2$ near ground level, while air masses heavily influenced by photosynthetic uptake (with less $CO_2$) and coming from the outer domain play a key role at higher altitudes, especially on 22 August. This could explain the dip seen on this date (see the pink box in Fig. R1). The animation of biogenic concentrations over D01 attached in the supplement provides a visual perspective of this phenomenon.

[Figure]

Figure R1: Time series of the daily mean measured values over five sites of MUCCnet (black) and the averaged modeled $XCO_2$ from CAMS (red, CAMS.$XCO_2$) and WRF over D03 during the daytime (i.e., 6:00 UTC to 17:00 UTC). The modelled column concentrations are pressure-weighted means (see Eq. 1 in the manuscript). The error bars represent the standard deviation of the simulated values over D03 and the measured over the five sites of MUCCnet. The orange curve represents the mean modelled column background concentration (Model.$XCO_2$_BCK). The green curve shows the averaged total column concentration (Model.$XCO_2$_BCK+Model.$XCO_2$_ANT+Model.$XCO_2$_BIO) and the blue curve shows the averaged column concentrations considering only the background and anthropogenic activities (Model.$XCO_2$_BCK+Model.$XCO_2$_ANT), without biospheric fluxes.

Owing to the relatively large bias of $CO_2$ brought in by CAMS, we considered using the model-measurement MB over all the measurement dates (i.e., 3.7 ppm) to "correct" the modelled values. This could help to see if the model could reproduce similar variations to those seen by the measurements. These variations are determined by the modelled biogenic effects, initial emission fluxes from the inventory, the modelled advection of air masses influenced by human and biogenic fluxes, etc.

However, this is not the case for $CH_4$, since no significant model-measurement bias can be found in the daily-mean $XCH_4$ (cf. Fig. 4(c) & (d) of the updated manuscript). Due to the quite weak biogenic activities of $CH_4$ in and around Munich (cf. Fig. 6 of the updated manuscript), the model-measurement bias of $CH_4$ is mostly caused by the uncertainties in human-related emissions.

To eliminate the $CO_2$ bias which could be mostly caused by the overestimation of background and to better observe the day-by-day variations in $XCO_2$ (see Fig. M4), we subtract the MB over the entire available measurement period (i.e., the 15 dates shown in Fig. M4; 3.7 ppm) from the modelled values for all sites and for each

available measurement date. In the manuscript, we chose to show 7 continuous days (from 16 to 22 August) as our key study period (see Sect. M4.3.2), while the rest is included in the supplement (see Sect. S7). Therefore, the MB discussed in of Sect. M4.2.3 is the remaining mean bias over these 7 consecutive days, after correcting with the value derived from the full 15 days.

[Figure]

Figure R2: Vertical profiles of (a)Altitudes, averaged modelled $CO_2$ over D03 on (b)16, (c)17, and (d)22, August 2018 from CAMS and WRF-Chem. The red curve represents the values from CAMS, and the others stand for our model results, with green for the total values, blue for the sum of the background and the human-related enhancements, and red for the background.

To clearly state this cancellation in Sect. 4.2.3, we have added the following sentence:

| Lines 331∼335 | *As described in Sect. 4.2.2., a MB of 3.7 ppm in $CO_2$ has been found over all the available measurement dates (see Fig. 4), which is defined to be the difference between the smoothed and measured daily mean $XCO_2$ and the modelled values. To eliminate the bias (too high modelled background $CO_2$) and focus on the model-measurement differences due to other causes, this MB is subtracted from the modelled $XCO_2$ in the day-by-day model-measurement comparison for all sites and for each simulation date.* |
|---|---|

2. Was CO measured with the EM27 as this would provide an independent constraint? If not then this should be mentioned as an additional valuable data to collect as new EM27's can do this together with CO2 and CH4.

Response: Thank you for this comment. CO is also measured by MUCCnet, but we did not include it in our modeling framework, and is outside the scope of the current study. In an ongoing project, we consider studying CO in the following step, and we have added this information in the conclusion and outlook of this study.

| Lines 595∼596 | *A study into the use of the simultaneously measured total column carbon monoxide (XCO) to constrain emissions from combustion processes can be carried out.* |
|---|---|

3. For methane the EM27 measures the total column, including the stratosphere where it falls off. TCCON does correct for this using HF that unfortunately the EM27 does not measure. The gradient method and analysis

assumes this is constant and this should be clearly stated with citations (Saad et al). If this correction is not made the observations should be biased a low.

Response: Thanks you for this valuable suggestion and the recommendation of Saad's study. In this study, the HF correction was not implemented in the retrieval process of EM27/SUNs in MUCCnet. The HF correction is applied to correct the tropopause heights of the a-priori $CH_4$ profiles used in the retrieval and the effect of this corrections on the $XCH_4$ coefficient is verified to be small and well within the error bars (Geibel et al., 2012). Many thanks for your recommendations and we have added the information of this correction to the content.

| Lines 236 | *Additionally, the hydrogen fluoride (HF) correction (Saad et al., 2014) is not applied in our retrieval process of $CH_4$.* |
|---|---|

The authors find a slight +ve bias "while in general the observed values are slightly higher, with a linear regression slope of 0.73 and a negative MB (-1.8 ± 4.0 ppb). This small bias could be caused by the initial and lateral boundary conditions from CAMS, or due to unknown or underestimated emissions" The possible reasons for this should be explained more clearly.

Response: From the perspective of the model itself, the bias in $CH_4$ concentrations is mainly attributable to the uncertainties in human activities. Bottom-up multinational emission inventories of $CH_4$ are generally compiled by scaling emissions using activity data and emission factors, which results in relatively large uncertainties (Bergamaschi et al., 2022). For the TNO_GHGco_v1.1 emission inventory used in this study, its point source information was collected on the location of power plants, large industrial installations, oil and gas production sites, airports and waste treatment locations (e.g. landfills), mostly from the E-PRTR (European Pollutant and Transfer Register) database. The data are valid for 2015. Therefore, in addition to the uncertainties due to the quantification of emissions in the inventory mentioned above, inconsistencies between emission information collected in 2015 or even earlier and actual emissions during the study period in 2018, could result in differences. Despite having chosen a high-resolution, state-of-the-art emission inventory, these uncertainties could contribute to the model-measurement differences. We have added the discussion related to causes of the uncertainties in the inventory to the manuscript as follows,

| Lines 550∼557 | *In this study, the modelled contributions from human activities are initialized with the emission fluxes from the emission inventory TNO_GHGco_v1.1 for the year 2015. The multinational bottom-up emission inventory holds large uncertainties, due to the large variability in spatiotemporal distributions of $CH_4$ emissions from different sectors in different regions that have not yet been fully captured by the emission inventory (Bergamaschi et al., 2022), the disaggregation from annual emissions to hourly values using temporal profiles and the temporal inconsistency of emission information from 2015 or even earlier than the study period in 2018. This could result in missing or underestimated emissions in the inventory, as suggested by the measurements. After delineating the areas where the uncertain sources could be located, they were further pinpointed based on the updated database and local knowledge.* |
|---|---|

In addition to the uncertainties in the human emissions, we also extended our discussion of other potential causes in the $CH_4$ biases. The modelled meteorological fields could bring about errors in the advection, which would contribute to the biases of absolute methane concentrations ($XCH_4$) and further to discrepancies in the methane gradients. Furthermore, due to lags in the time it takes for air to reach the boundary, the variation of the background signals at each instrument could be large. This is definitely a key point to be considered when the concentration gradients are used for inversions to optimize the inventories (as in Jones et al., 2021), but it presents significant complexities for our study and its implementation within the WRF-based framework. In addition to the emission-related causes that lead to the model-measurement biases of concentrations and their gradients, Hedilius et al.(2017) pointed out that non-emission factors (like the mixed layer height and

topography) would further cause biased results. The importance of topography is verified as a significant factor in the variations of concentrations beyond the urban area. In our case, even though the elevations over our innermost domain (DO3, Munich) are rather consistent, i.e. around 550 m above sea level, the area around this domain's boundaries contains the complex topography of the Alps. It should be noted, that while it still plays a role, column measurements are less sensitive to mixing layer height than are in-situ measurements. The causes discussed here could contribute to errors in the concentration and thus, the gradients. We have extended the discussion other causes which could contribute to the biases in the model-measurement gradients as such:

| | |
|---|---|
| Lines 540~544 | *In addition to the errors caused by the uncertainties in the initial emission inventory, other potential causes could contribute to errors in the concentration and thus, the gradients as well. The bias brought by the modelled meteorological fields can contribute to the bias of the modelled $XCH_4$, further to discrepancies in $\Delta XCH^S_{4,sla}$, by influencing on the advection (Wu et al., 2017). The variations in background concentrations at each site due to the lags in time when it takes for the in-flowing air to reach the boundary (as discussed in Jones et al. (2021)). Moreover, the non-emission factors (e.g., the mixed layer height and topography) could also introduce biased results results (Hedelius et al., 2017).* |

4. There are many EM27 model studies of optimized fluxes such as Jones et al, Viatte et al that are cited. Another very relevant study Heerah et al JGR Atm 2022 that uses distributed EM27 data and WFR model to do systematic comparison with winds and inverse modeling for dairies should also be cited.

Response: Thank you for this recommendation. It is indeed a relevant study, and we have included a reference to it in the manuscript as follows:

| | |
|---|---|
| Lines 86~89 | *For instance, highly-resolved meteorological fields can drive particle transport models (Fasoli et al., 2018). These Lagrangian footprints can then be used for inversion studies, similar to Heerah et al. (2021), who optimized dairy $CH_4$ emissions across the San Joaquin Valley using WRF-STILT inversions. Currently, an adapted Bayesian inversion model based on Jones et al. (2021) is being developed to infer anthropogenic $CO_2$ emissions, with the consideration of biogenic fluxes.* |

**References**

Bergamaschi, P., Segers, A., Brunner, D., Haussaire, J.-M., Henne, S., Ramonet, M., Arnold, T., Biermann, T., Chen, H., Conil, S., Delmotte, M., Forster, G., Frumau, A., Kubistin, D., Lan, X., Leuenberger, M., Lindauer, M., Lopez, M., Manca, G., Müller-Williams, J., O'Doherty, S., Scheeren, B., Steinbacher, M., Trisolino, P., Vítková, G., and Yver Kwok, C.: High-resolution inverse modelling of European $CH_4$ emissions using the novel FLEXPART-COSMO TM5 4DVAR inverse modelling system, Atmos. Chem. Phys., 22, 13243–13268, 10.5194/acp-22-13243-2022, 2022.

Heerah, S., Frausto-Vicencio, I., Jeong, S., Marklein, A. R., Ding, Y., Meyer, A. G., Parker, H.A., Fischer, M.L., Franklin, J.E., Hopkins, F.M. and Dubey, M. Dairy methane emissions in California's San Joaquin Valley inferred with ground-based remote sensing observations in the summer and winter. Journal of Geophysical Research: Atmospheres, 126(24), e2021JD034785, 10.1029/2021JD034785, 2021.

Hedelius, J.K., Feng, S., Roehl, C.M., Wunch, D., Hillyard, P.W., Podolske, J.R., Iraci, L.T., Patarasuk, R., Rao, P., O'Keeffe, D. and Gurney, K.R.: Emissions and topographic effects on column $CO_2$ ($XCO_2$) variations, with a focus on the Southern California Megacity. Journal of Geophysical Research: Atmospheres, 122(13), pp.7200-7215, 10.1002/2017jd026455, 2017.

Geibel, M. C., Messerschmidt, J., Gerbig, C., Blumenstock, T., Chen, H., Hase, F., Kolle, O., Lavrič, J. V., Notholt, J., Palm, M., Rettinger, M., Schmidt, M., Sussmann, R., Warneke, T., and Feist, D. G.: Calibration of column-averaged $CH_4$ over European TCCON FTS sites with airborne in-situ measurements, Atmos. Chem. Phys., 12, 8763–8775, 10.5194/acp-12-8763-2012, 2012.

Jones, T. S., Franklin, J. E., Chen, J., Dietrich, F., Hajny, K. D., Paetzold, J. C., Wenzel, A., Gately, C., Gottlieb, E., Parker, H., Dubey, M., Hase, F., Shepson, P. B., Mielke, L. H., and Wofsy, S. C.: Assessing urban methane emissions using column-observing portable Fourier transform infrared (FTIR) spectrometers and a novel Bayesian inversion framework, Atmos. Chem. Phys., 21, 13131–13147, 10.5194/acp-21-13131-2021, 2021.

Viatte, C., Lauvaux, T., Hedelius, J. K., Parker, H., Chen, J., Jones, T., Franklin, J. E., Deng, A. J., Gaudet, B., Verhulst, K., Duren, R., Wunch, D., Roehl, C., Dubey, M. K., Wofsy, S., and Wennberg, P. O.: Methane emissions from dairies in the Los Angeles Basin, Atmos. Chem. Phys., 17, 7509–7528, 10.5194/acp-17-7509-2017, 2017.

Wunch, D., Toon, G. C., Sherlock, V., Deutscher, N. M., Liu, C., Feist, D. G., and Wennberg, P. O.: Documentation for the 2014 TCCON Data Release, 10.14291/TCCON.GGG2014.DOCUMENTATION.R0/1221662, 2015.

Wu, D., Lin, J. C., Fasoli, B., Oda, T., Ye, X., Lauvaux, T., Yang, E. G., and Kort, E. A.: A Lagrangian approach towards extracting signals of urban $CO_2$ emissions from satellite observations of atmospheric column $CO_2$ ($XCO_2$): X-Stochastic Time-Inverted Lagrangian Transport model ("X-STILT v1"), Geosci. Model Dev., 11, 4843–4871, 10.5194/gmd-11-4843-2018, 2018.

Saad, K. M., Wunch, D., Toon, G. C., Bernath, P., Boone, C., Connor, B., Deutscher, N. M., Griffith, D. W. T., Kivi, R., Notholt, J., Roehl, C., Schneider, M., Sherlock, V., and Wennberg, P. O.: Derivation of tropospheric methane from TCCON $CH_4$ and HF total column observations, Atmos. Meas. Tech., 7, 2907–2918, 10.5194/amt-7-2907-2014, 2014.